# Structure of the proteolytic enzyme PAPP-A with the endogenous inhibitor stanniocalcin-2 reveals its inhibitory mechanism

Sara Dam Kobberø [1,6], Michael Gajhede [2,6], Osman Asghar Mirza[2], Søren Kløverpris[1,4], Troels Rønn Kjær [1,5], Jakob Hauge Mikkelsen[1,5], Thomas Boesen [3] & Claus Oxvig [1] ✉

The metzincin metalloproteinase PAPP-A plays a key role in the regulation of insulin-like growth factor (IGF) signaling by specific cleavage of inhibitory IGF binding proteins (IGFBPs). Using single-particle cryo-electron microscopy (cryo-EM), we here report the structure of PAPP-A in complex with its endogenous inhibitor, stanniocalcin-2 (STC2), neither of which have been reported before. The highest resolution (3.1 Å) was obtained for the STC2 subunit and the N-terminal approximately 1000 residues of the PAPP-A subunit. The 500 kDa 2:2 PAPP-A·STC2 complex is a flexible multidomain ensemble with numerous interdomain contacts. In particular, a specific disulfide bond between the subunits of STC2 and PAPP-A prevents dissociation, and interactions between STC2 and a module located in the very C-terminal end of the PAPP-A subunit prevent binding of its main substrate, IGFBP-4. While devoid of activity towards IGFBP-4, the active site cleft of the catalytic domain is accessible in the inhibited PAPP-A·STC2 complex, as shown by its ability to hydrolyze a synthetic peptide derived from IGFBP-4. Relevant to multiple human pathologies, this unusual mechanism of proteolytic inhibition may support the development of specific pharmaceutical agents, by which IGF signaling can be indirectly modulated.

Although first discovered as a placental antigen present in the circulation during human pregnancy, the metzincin metalloproteinase PAPP-A (pregnancy-associated plasma protein-A, pappalysin-1, EC 3.4.24.79) is ubiquitously expressed in human tissues[1]. PAPP-A is able to cleave insulin-like growth factor binding proteins (IGFBP) -2, -4, and -5 in the extracellular environment. It consequently causes the release of bound bioactive IGF-1 or -2, which are potent growth factors[2–4]. Together with PAPP-A2[5], PAPP-A is a key regulator of IGF signaling[6]. This is illustrated by the PAPP-A knockout mouse, which is a proportional dwarf reduced ~40% in size[7]. Apart from growth physiology,

PAPP-A is believed to be involved in the pathogenesis of multiple human age-related diseases[8].

The defining feature of the metzincin superfamily of metalloproteinases is the presence of the extended zinc-binding (HEXXHXXGXXH/D) and Met-turn motifs in the catalytic domain (CD)[9–11], widely known from the large family of matrix metalloproteinases (MMPs). The multimodular PAPP-A (Fig. 1) is the founding member of the pappalysin family[12], which includes PAPP-A2 as the only paralogue. PAPP-A and PAPP-A2 have orthologues, at least in vertebrates, and distant family members with which only

[1]Department of Molecular Biology and Genetics, Aarhus University, DK-8000 Aarhus C, Denmark. [2]Department of Drug Design and Pharmacology, University of Copenhagen, DK-2100 Copenhagen Ø, Denmark. [3]Interdisciplinary Nanoscience Center, Aarhus University, DK-8000 Aarhus C, Denmark. [4]Present address: Agilent Technologies, DK-2600 Glostrup, Denmark. [5]Present address: Department of Biomedicine, Aarhus University, DK-8000 Aarhus C, Denmark. [6]These authors contributed equally: Sara Dam Kobberø, Michael Gajhede. ✉e-mail: co@mbg.au.dk

**Fig. 1 | Modular structure of the 1546-residue PAPP-A (pappalysin-1) subunit.** Approximate domain boundaries and the cysteine residue responsible for PAPP-A dimerization are indicated. Note that two Lin12-Notch repeat (LNR) modules (LNR1-2) are inserted into the catalytic domain (CD), and a third (LNR3) is present as part of the C domain. No sequence similarity has been detected for the central region of

500 residues, which is designated M. The short consensus repeats (SCR) are also known as CCP domains. PAPP-A promotes IGF signaling by binding via SCR3-4 to glycosaminoglycan (GAG), present at the cell surface. This allows proteolytic release of IGF in close proximity to the IGF receptor.

the CD is shared. Among the latter, referred to as unicellular pappalysins[13], the archaeal proteinase ulilysin (29 kDa)[14] from *M. acetivorans* is the best characterized, and together with mirolysin (31 kDa)[13] from *T. forsythia*[13], the only members of the pappalysin family with known three-dimensional structures.

In addition to the CD, the N-terminal third of PAPP-A harbors a laminin G-like (LG) domain of ~250 residues with weak sequence similarity to, e.g., the α-chain of laminins and sex hormone-binding globulin[15]. No function has been assigned to this domain. The C-terminal fourth of PAPP-A contains five short consensus repeat (SCR) modules, also known as complement control protein (CCP) or sushi domains. These domains (SCR3 and SCR4) bind cell surface glycosaminoglycans (GAGs), hereby securing the release of IGF in close proximity to the IGF receptor (IGF1R)[16,17]. The middle region (M) of the PAPP-A subunit between the CD and SCR1 is poorly characterized, but a sequence stretch in the C-terminal end of this region is known to be required for dimerization[18]. Two PAPP-A subunits of 1546 residues form a disulfide-bound homodimer of ~400 kDa via C1210, which is located just N-terminal to SCR1.

Finally, the PAPP-A subunit contains three Lin12-Notch repeat (LNR) modules, two of which (LNR1-2) are inserted into the CD, and a third (LNR3), which is located C-terminal to SCR5[19] (Fig. 1). Each of the PAPP-A LNR modules are known to bind a Ca²⁺ ion, and disruption of Ca²⁺ ion binding by single amino acid substitution in either of the LNR modules causes a complete loss of proteolytic activity toward IGFBP-4, while cleavage of IGFBP-5 is unaffected[19].

The paralogue proteins stanniocalcin (STC)-1 and -2 were recently identified as potent endogenous PAPP-A inhibitors and thus connected with the IGF system[20,21]. Both STCs are dimers of ~250-residue subunits, which do not share sequence similarity with any other human proteins. Except for PAPP-A2, no other target proteinases of the STCs have been identified[20]. STC1 is a competitive PAPP-A inhibitor ($K_i = 68$ pM)[20], while STC2 inhibits irreversibly via a disulfide bond formed between an unknown cysteine residue of PAPP-A and C120 of STC2, which has no counterpart in STC1[21]. Common metzincin inhibitors do not inhibit PAPP-A, but PAPP-A circulates during human pregnancy at high levels as an inactive complex with the proform of eosinophil major basic protein (proMBP), also synthesized in the placenta[22].

Accumulating data support a model in which the balance between active and STC-inhibited PAPP-A is a major determinant of IGF signaling in tissues[21,23,24]. For example, single amino acid STC2 variants with reduced PAPP-A inhibitory activity cause a dramatic increase in adult human height[25]. Recognition of the physiological roles of these components has led to the concept of a regulatory STC2 → PAPP-A → IGFBP → IGF axis[1].

Structural studies of PAPP-A have so far been unsuccessful, possibly caused by inherent flexibility of the protein, a high degree of glycosylation[26], and by difficulties in the expression of isolated domains[15]. No structural information is available for the STCs.

In this work, we present the structure of PAPP-A in complex with STC2 based on cryo-electron microscopy (cryo-EM). We find that the 2:2 PAPP-A·STC2 complex of ~500 kDa is a highly flexible structure with multiple contacts within and between the subunits. Disulfide bonds link the two PAPP-A subunits and the two STC2 subunits, and a disulfide bond connecting each PAPP-A subunit with an STC2 subunit prevents dissociation of the complex. Although the inhibited complex cannot cleave IGFBP-4, the PAPP-A active site is accessible and active toward a synthetic peptide derived from IGFBP-4.

## Results

### Experimental strategy for structural analysis of the PAPP-A·STC2 complex

Recombinant PAPP-A·STC2 complex was purified following co-culture of mammalian cell lines secreting active site-inactivated PAPP-A (E563Q) or wild-type STC2 (Fig. 2a). A 2:2 stoichiometry of the PAPP-A·STC2 complex was confirmed by mass photometry, the integrity of the polypeptides was assessed by mass spectrometry (Supplementary Fig. 1), and the complex was subjected to single-particle cryo-EM data collection (Supplementary Fig. 2) for subsequent structural analysis. The first processing of the data yielded a 3.1 Å resolution map (MAP1, Supplementary Fig. 2c, d and 3a, and Table 1) that showed well-defined density in the LG, CD, M, and SCR2 domain regions of one of the PAPP-A subunits as well as for residues 44–210 of one of the STC2 subunits. This corresponds to ~36% of the entire 2:2 complex or ~73% of the PAPP-A momomer (Supplementary Table 1 and Supplementary Fig. 4). Only weak traces of density of the other PAPP-A and STC2 subunits were observed, indicating pronounced particle flexibility. Two 2-fold symmetric density maps covering the entire 2:2 PAPP-A·STC2 complex with overall resolutions of 4.0 Å (MAP2) and 5.0 Å (MAP3) were subsequently generated (Supplementary Figs. 2e, f and 3c, d). The resolution of MAP2 varies between 3 and 8 Å (regions with the lowest resolution is highlighted in Supplementary Fig. 5). Variability analysis[27] of this map revealed that the PAPP-A·STC2 dimer is indeed a highly flexible structure (Supplementary Fig. 6). MAP2-showed well-defined density but with streaky features in the PAPP-A dimer interface. These were not present in MAP3, which was used to build this region.

Subsequent model building was based on initial monomeric models calculated using AlphaFold2 (AF2)[28], as the program in our hands failed to generate models comprising PAPP-A dimers. For the PAPP-A monomer, a clear correlation between resolution and residue estimate of confidence was not obvious, however, low-resolution domains (M5 and SCR4) also display low confidence levels. This is probably reflecting that the resolution achieved is primarily determined by the flexibility of the full complex, and that these domains are the primary mediators of the flexibility. For the STC2 monomer, the correlation was clear (Supplementary Fig. 7). There is good agreement

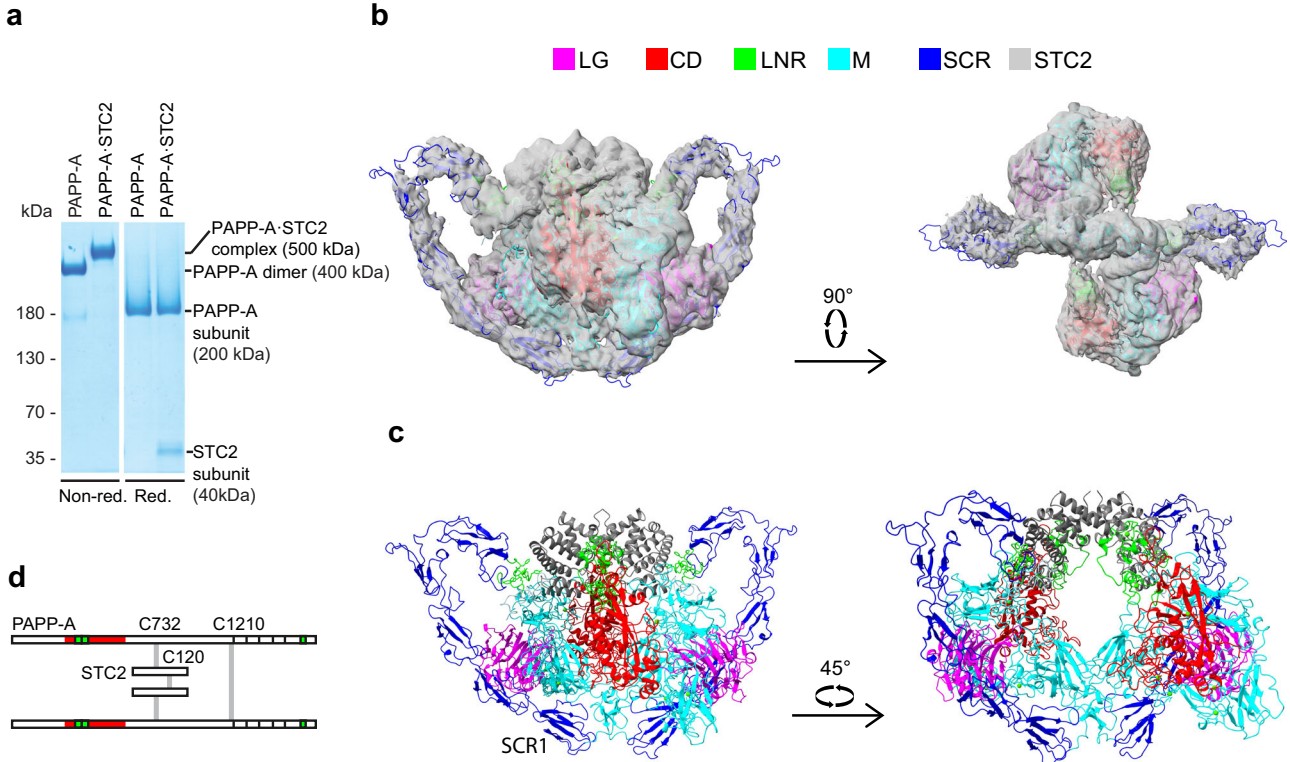

**Fig. 2 | The three-dimensional structure of the human PAPP-A·STC2 complex.**
**a** SDS-PAGE separation of purified PAPP-A dimer and PAPP-A·STC2 complex used for cryo-EM analysis. Samples were reduced or non-reduced, as indicated, and the gel was stained with Coomassie brilliant blue. Active site-inactivated PAPP-A (E563Q) was used for this experiment and for further structural studies. The experiment was repeated >10 times with similar results. Source data are provided as a Source data file. **b** The experimental volume is shown with the final model of the 2:2 PAPP-A·STC2 complex in two orientations. Domains are colored as indicated. This color scheme is used throughout the figure. The overall approximate dimensions (W × H × D) of the PAPP-A·STC2 complex are 200 × 115 × 175 Å. **c** Cartoon representation of the refined PAPP-A·STC2 model in two orientations. The map (MAP2) was contoured at 4.5σ. **d** Schematic diagram of the covalent, heterotetrameric PAPP-A·STC2 2:2 complex, highlighting interchain cysteine connectivity.

in the STC2 regions predicted to be folded by AF2, and accordingly, the regions predicted to be intrinsically disordered (residues 24–43 and 211–302) were not observed in the experimental structure. The predicted and the experimental structures of STC2 superimpose well in the folded regions. The most obvious difference between the PAPP-A monomer resulting from AF2 calculations and the PAPP-A subunit of the 2:2 experimental PAPP-A·STC2 complex is the markedly different spatial directions of the SCR modules (Supplementary Fig. 7c).

### Overall structure of the PAPP-A·STC2 complex
Our results show that the PAPP-A dimer binds a single STC2 dimer in a 2:2 complex with many intra- and inter-subunit contacts present in the structure (Fig. 2b and Supplementary Fig. 8). The PAPP-A subunit is dominated by β-strands while the STC2 subunit adopts an all-helical conformation. The 10 SCR modules form a wide heart-shaped figure with the SCR1 modules on either side of the apex and the STC2 dimer at the top. Most of the remaining domains are symmetrically arranged on each side of the plane of the SCR domains with the active site clefts of the CDs facing inward toward a central cavity (Fig. 2c and Supplementary Fig. 9).

The PAPP-A and STC2 subunits contain 82 and 14 cysteine residues, respectively. The connectivity of the majority of these was determined experimentally, including the four interchain disulfide bridges by which the 2:2 PAPP-A·STC2 complex is held together (Fig. 2d and Supplementary Fig. 10). Of the 14 potential N-glycosylation sites of the PAPP-A subunit, eight are occupied (N390, N402, N429, N480, N601, N725, N825, and N1323). The single potential N-glycosylation site of STC2 is not occupied (Supplementary Fig. 11).

### The N-terminal laminin G-like (LG) domain
The LG domain (E82-P324) is a β-sandwich principally composed of 15 antiparallel β-strands arranged in two sheets, similar to other members of the LG family, and an α-helical segment of two turns below one of the sheets (Fig. 3). The domain contains two cysteine residues which form a disulfide bond (C144-C235) with no counterpart in other LG domains. Some LG domains bind $Ca^{2+}$ ions[29], but the ligating residues are not conserved in PAPP-A[15], and we observe no density in the map to suggest $Ca^{2+}$ ion binding of this domain. Remarkably, although exon 2 of the PAPPA gene encodes a large portion of the N-terminal third of PAPP-A (G139-R493)[30], the LG and the CD (P325-K671) domains do not make contact, but rather lie on each side of a J-shaped structure formed by domains of the M region (P672-D1214) of the same subunit (Fig. 3).

### The catalytic domain (CD)
The core structure of the CD (P325-K671) is composed of a twisted β-sheet of four parallel strands on top of four α-helices (Fig. 4a). Despite low sequence similarity, the fold of this core is similar to the core of the two structurally characterized pappalysins, as well as other metzincins (Supplementary Fig. 12a). The 14 cysteine residues of the CD form six disulfide bonds, one of which (C457-C473) could not be experimentally verified in the map, while two cysteine residues (C461 and C600) are unpaired (Supplementary Fig. 10).

The active site cleft of the CD is present in front of a central α-helix under the edge of the β-sheet, which also provides two of the three zinc-coordinating histidine residues. The canonical methionine residue of the Met-turn, known to be critical for PAPP-A activity[12], is

**Table 1 | CryoEM data collection, refinement, and validation statistics**

| DATA COLLECTION AND IMAGE PROCESSING | MAP 1 | MAP 2 | MAP 3 |
|---|---|---|---|
| Microscope | Titan Krios G3i | Titan Krios G3i | Titan Krios G3i |
| Energy filter | Bioquantum | Bioquantum | Bioquantum |
| Detector | Gatan K3 | Gatan K3 | Gatan K3 |
| Magnification | 165,000 | 165,000 | 165,000 |
| Voltage (kV) | 300 | 300 | 300 |
| Cumulative dose (e.Å$^{-2}$) | ~59 and ~58 | ~59 | ~59 |
| Defocus range (μm) | 0.6–1.8 | 0.6–1.8 | 0.6–1.8 |
| Pixel size (Å) | 0.507 | 0.507 | 0.507 |
| Motion correction | cryoSPARC v3 | cryoSPARC v3 | cryoSPARC v3 |
| CTF estimation | cryoSPARC v3 | cryoSPARC v3 | cryoSPARC v3 |
| Particle selection | cryoSPARC v3 | cryoSPARC v3 | cryoSPARC v3 |
| 3D classification and refinement | cryoSPARC v3 | cryoSPARC v3 | cryoSPARC v3 |
| Micrographs used | 33582 (Dataset I, 9854 and Dataset II, 23728) | 23728 (Dataset II) | 23728 (Dataset II) |
| Refinement symmetry | C1 | Combined C1 + C2 | Combined C1 + C2 |
| Picked particles | 6,677,370 | 601,179 | 601,179 |
| Particles for C2-expanded | | | 228,452 |
| Final particles | 278,982 | 114,226 | 31,119 |
| Map resolution (Å) | 3.06 | 4.05 | 5.02 |

| REFINEMENT | PARTIEL DIMER (MAP1) | FULL DIMER:DIMER (Composite MAP1 + MAP2 + MAP3) | |
|---|---|---|---|
| PDB ID | 8A7D | 8A7E | |
| EMDB ID | EMD-15220 | EMD-15221 (from EMD-15220, EMD-15217, EMD-15219) | |
| Composition: | | | |
| Chains | 3 | 4 | |
| Non-hydrogen atoms | 10,775 | 26,446 | |
| Protein residues | 1361 (921 + 167 + 273) | 3384 (336 + 3048) | |
| RMS deviations: | | | |
| Bond lengths (Å) | 0.003 | 0.002 | |
| Bond angles (°) | 0.611 | 0.541 | |

| VALIDATION | PARTIEL DIMER (MAP1) | FULL DIMER:DIMER (Composite MAP1 + MAP2 + MAP3) | |
|---|---|---|---|
| Molprobity score (percentile) Clashscore | 7.83 | 16.34 | |
| Ramachandran plot: | | | |
| Outliers (%) | 0.44 | 0.12 | |
| Allowed (%) | 5.25 | 5.81 | |
| Favored (%) | 94.30 | 94.08 | |
| Ramachandran plot Z-score RMSD: | | | |
| Whole | −0.22 (0.24), ($N$ = 1352) | −0.32 (0.15), ($N$ = 3376) | |
| Helix | 0.17 (0.33), ($N$ = 239) | 0.30 (0.24), ($N$ = 496) | |
| Sheet | −0.83 (0.33), ($N$ = 251) | −0.67 (0.20), ($N$ = 680) | |
| Loop | 0.10 (0.23), ($N$ = 862) | −0.09 (0.15), ($N$ = 2200) | |
| Rotamer outliers (%) | 0.50 | 0.0 | |
| Cβ deviations (%) | 0.0 | 0.0 | |
| Peptide plane (%): | | | |
| Cis-proline | 7.5 | 7.0 | |
| Cis-general | 0.0 | 0.06 | |
| Twisted proline | 1.25 | 0.0 | |
| Twisted general | 0.0 | 0.0 | |
| CaBLAM outliers (%) | 2.53 | 2.68 | |
| ADP (B-factors): | | | |
| Iso/Aniso (#) | 10771/0 | 26442/0 | |
| Min/max/mean: | | | |
| Protein | 0.00/215.19/100.97 | 97.63/747.84/270.48 | |
| Ligand | 30.00/185.77/35.67 | 117.96/400.99/265.28 | |
| Model vs. data: | | | |
| CC_mask | 0.7860 | 0.7818 | |
| CC_volume | 0.7765 | 0.7665 | |
| CC_peaks | 0.6078 | 0.7315 | |

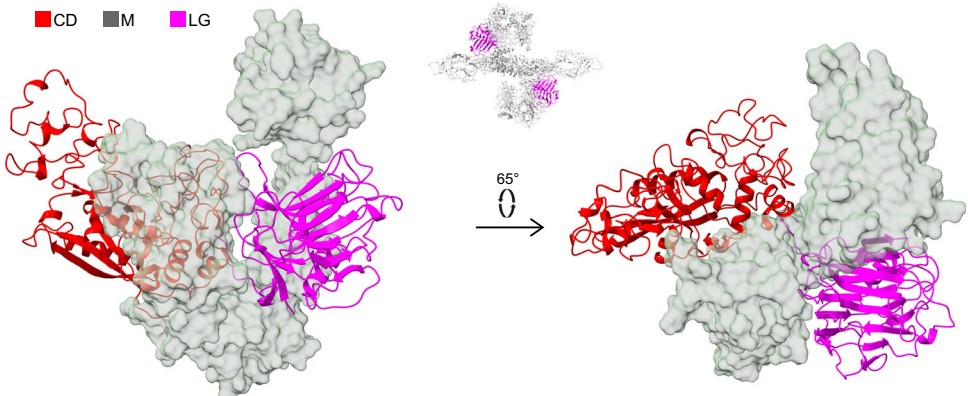

**Fig. 3 | The laminin G-like (LG) domain of the PAPP-A subunit.** Representation to emphasize the separation of the CD (red, cartoon) and the LG domain (magenta, cartoon) by the M domains (transparent surface) in two different orientations. The position of the LG domain in the tetrameric PAPP-A·STC2 structure is shown with the inset above the arrow (top view as in Fig. 2b).

positioned in the active site cleft, where it is responsible for maintaining the zinc-binding geometry[31] (Fig. 4b). This active site environment of PAPP-A is highly similar to that of ulilysin, and also other representative members of the metzincin superfamily (Supplementary Fig. 12b).

The 63-residue sequence stretch between the zinc-binding site and the Met-turn methionine forms a double-loop structure (the ZnM loop, V573-F635), which lies as an ~20 Å extension of the lower part of the active site cleft. The ZnM loop contains one of the two unpaired cysteines, and it is stabilized by three disulfide bonds, two of which pair with cysteine residues outside the loop (Fig. 4b). The observed map density is in agreement with the coordination of a $Ca^{2+}$ ion in the N-terminal end of the ZnM loop (Fig. 4b). The two inserted LNR modules (see below) are present on the same face of the CD, but lie above the active site cleft (Fig. 4a).

Previous data have demonstrated that the PAPP-A·STC2 complex is completely inactive toward IGFBP-4[21], the main substrate of PAPP-A[32], but surprisingly, the structure reveals that active site cleft is not occupied. Concordantly, a synthetic 26-residue peptide derived from IGFBP-4 (L(118)QKHFAKIRDRSTSGGKMKVNGAPRE(143)), which spans the scissile bond, can be hydrolyzed by PAPP-A·STC2 (Fig. 4c). Although the catalytic efficiency is reduced compared to the uninhibited PAPP-A dimer, this still demonstrates that the catalytic machinery within the complex is functional.

## The central region (M) of the PAPP-A subunit
The M region folds into three large antiparallel β-sandwich structures, M1 (P672-F701 + L884-G943), M2 (F702-P883, loops out from M1), and M5 (F1019-F1181) (Fig. 5a). Three smaller domains of coils and strands separate M2 from M5 (M3, S944-G995 and M4, D996-G1018), and M5 from SCR1 (M6, D1182-D1214), respectively. The LG domain lies embedded in the J-shaped structure formed by M1-5 and is separated from the CD by more than 10 Å (Fig. 3). M2 has long inter-strand loops facing the side of the active site cleft (Fig. 5b). Based on map density and in agreement with the position of potentially coordinating acidic residues, we find evidence of one bound $Ca^{2+}$ ion in domains M2 and M4, and two in M3 (Supplementary Fig. 13a, b).

The M region contains 24 cysteine residues in total, and each M domain contains 1–4 intradomain disulfide bridges, except for M1 (Supplementary Fig. 10). C732 of M2 forms an interchain disulfide bond with C120 of STC2 (Fig. 5c). Concordantly, the PAPP-A variant C732A cannot form a covalent complex with STC2 (Fig. 5d), demonstrating that C732 of PAPP-A is the only cysteine residue responsible for specific covalent linkage of the dimers within the complex.

PAPP-A intersubunit contacts are formed between a 10-residue loop (P1147-Y1157) of M5 and all of M6, which also contains the single PAPP-A dimerization disulfide bond (C1210-C1210). The two M6 modules comprise a short α-helix followed by an antiparallel β-sheet and together form a small globular dimerization domain with four strands in the center. After the dimerization cysteine, the polypeptide continues toward the side of the LG domain of the opposite PAPP-A subunit (Fig. 5e).

The variability analysis suggests that the flexibility of the PAPP-A·STC2 complex arise from a continuous distribution of conformations of this domain (Supplementary Movie 1). This is possibly related to small energy differences between states of the M6 domain with a four-strand β-sheet and others with two separate two-strand sheets (Fig. 5e).

No sequence similarity has been reported for the domains M1 through M6. However, a structure-based search[33] revealed PFAM classification for M1 (fibronectin-iii type domain), and M2/5 (F5/8 type C or C2-like domain), the two latter M domains being structurally very similar (Supplementary Fig. 13c).

## The SCR domains
The SCR domains occupy the region of the PAPP-A·STC2 structure with the most pronounced flexibility, but the path of the five SCR modules (C1215-C1554) is evident with most of the domain axes deviating only slightly from the plane (Fig. 5f). SCR2 is structurally best resolved in the density map, probably because the LG, the CD, the M1-M5 domains, and the SCR2 domain together act as a rigid body. It adopts a β-sandwich structure composed of five antiparallel strands with three visible disulfide bonds, thus confirming structural similarity with the many known SCR structures[34]. The dimerization disulfide of M6 brings the N-termini of the SCR1 domains close to each other with an angle between their axes of ~90°. In each PAPP-A subunit, SCR1-2 lie on an almost straight line, and SCR3-5 turn progressively more inward, in particular SCR5, which makes a >100° turn downward toward the complex center (Fig. 5f). The M5 domain lies in the SCR plane, but M1-4 lie in a separate plane, which intersects the SCR plane (Fig. 5g).

From biophysical analyses of other SCR-containing proteins, it is established that residues at interdomain junctions of adjoining SCR domains can provide either flexibility or rigidity[34]. In the PAPP-A·STC2 complex, SCR1 is fixed and interacts with M5 in trans, SCR2 interacts with the LG domain in trans, and the C-terminal end of the PAPP-A subunit is also fixed (see below). In agreement with this, our data indicate pronounced flexibility of the SCR3-SCR5 region that does not

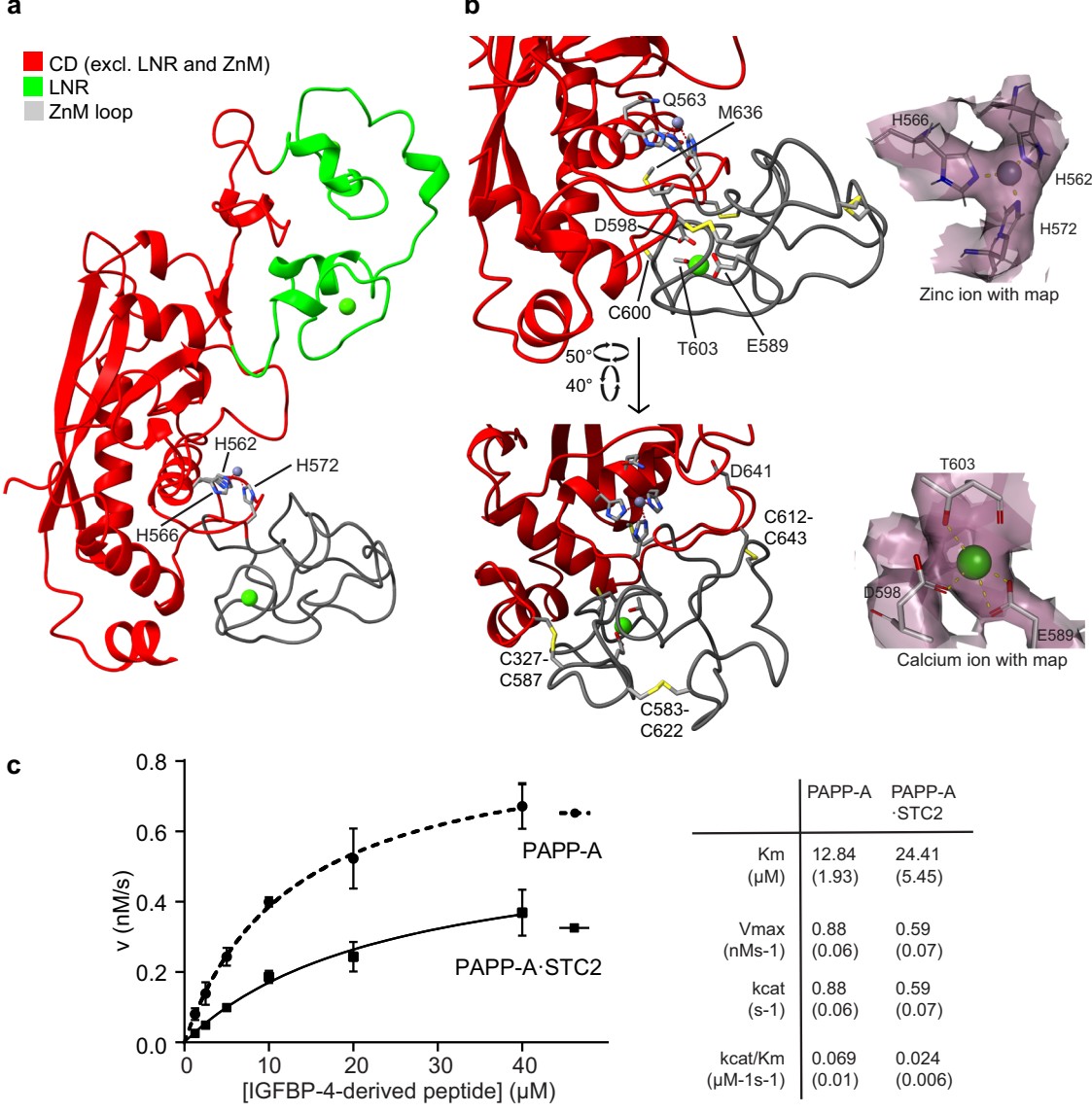

**Fig. 4 | The catalytic domain (CD) of the PAPP-A subunit. a** Cartoon representation of the isolated CD (red) with the ZnM loop (gray) and the two inserted LNR modules (LNR1-2, green) colored separately. This color scheme is used throughout the figure. The active site cleft, containing the Zn²⁺ ion coordinated by three histidine residues (H562, H566, H572), is facing right and located between the LNR modules and the ZnM loop. **b** Enlarged view of the active site cleft of the CD shown in two orientations. The methionine residue of the Met-turn (M636) is indicated. In the position of the glutamic acid residue (E563), which serves to polarize a water molecule involved in nucleophilic attack at the scissile peptide bond, is shown a glutamine residue, present in the recombinant protein (E563Q) used in this study. The ZnM loop is anchored to the core of the CD by two disulfide bonds (C327-C587

and C612-C643). One disulfide bond (C583-C622) and an unpaired cysteine residue (C600) is present within the ZnM loop. The position of a Ca²⁺ ion and coordinating residues (E589, D598, T603) of the ZnM loop are shown. Insets show map density around the active site Zn²⁺ ion (top) and the Ca²⁺ ion of the ZnM loop (bottom). The map (MAP1) was contoured at 4.0σ. **c** Dimeric PAPP-A and the 2:2 PAPP-A·STC2 complex are both able to cleave a 26-residue peptide derived from the PAPP-A substrate, IGFBP-4 (L(118)QKHFAKIRDRSTSGGKMKVNGAPRE(143), cleavage occurs at MK). The assay is based on intramolecular quenched fluorescence. Average values with error bars (SD) are plotted. Kinetic parameters with SD for both reactions are shown. n = 3 (PAPP-A) or 4 (PAPP-A·STC2) independent experiments. Source data are provided as a Source data file.

interact with other domains of the PAPP-A·STC2 complex. The SCR3 and SCR4 domains within this flexible region are responsible for GAG interaction, thus mediating PAPP-A cell surface binding[16].

## The LNR modules

LNR3 follows immediately after SCR5 and continues its downward direction to a position at level with LNR1-2 of the CD. The three ~30-residue LNR modules are similar to one another in structure with limited secondary structure in the form of short helical turns along a spiral-shaped path of the backbone, stabilized by disulfide bonds (Fig. 6a). Although Notch receptor LNR modules invariably contain three disulfides[35], the PAPP-A LNRs have either two or a single (Fig. 6b),

but the structural similarity is still evident (Supplementary Fig. 14). Residues of Notch LNRs known to participate in Ca²⁺ ion coordination are conserved in the LNRs of PAPP-A (Fig. 6b), and we observe map density corresponding to a Ca²⁺ ion in the space surrounded by the side chains of these residues, showing that a Ca²⁺ ion is also present in the PAPP-A LNR modules (Fig. 6c).

The very C-terminal residues following LNR3 (C1584-G1627) continue one turn of the LNR3 spiral and then points further downward. No map density was observed for the C-terminal 10 residues, in fair agreement with the lack of positive identification of the C-terminal eight residues of PAPP-A by mass spectrometry (Supplementary Fig. 1b). Except for a small two-strand sheet

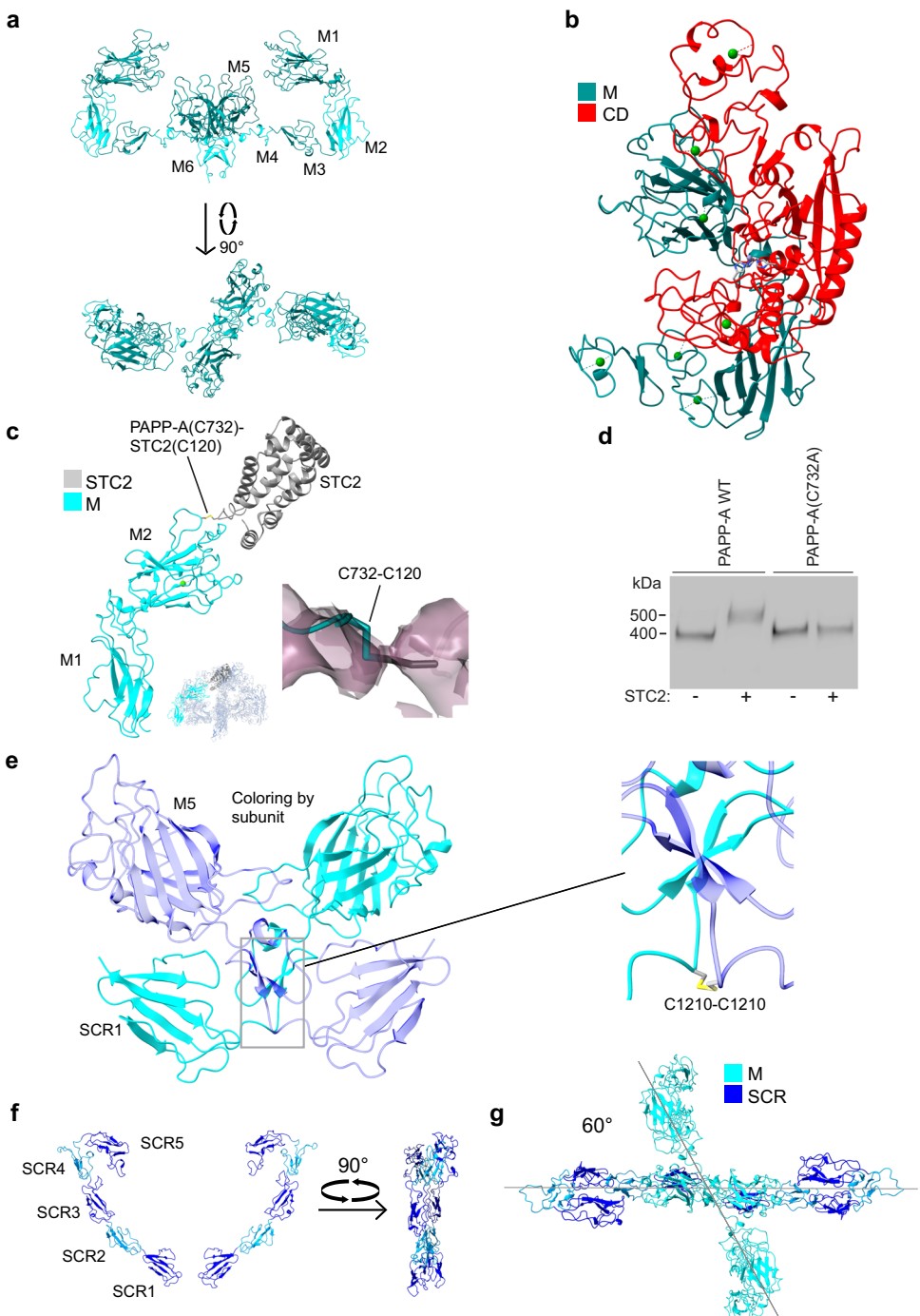

**Fig. 5 | Domains of the M region form a scaffold for the structure and lie in a plane with a dihedral angle of ~60° relative to the plane of the SCR domains.**
**a** Cartoon representation of the isolated M domains of the 2:2 PAPP-A·STC2 complex. The positions of the domains (M1 through M6) are indicated, and domain boundaries are shown by alternating shades of cyan. **b** Cartoon representation of the CD (red, active site cleft facing left) and domains M1-4 (cyan), showing the positions of inter-strand loops of M2 relative to the active site cleft within the PAPP-A·STC2 complex. **c** Cartoon representation showing the M1 and M2 domains, one STC2 subunit, and the disulfide bond between C732 of M2 and C120 of STC2, which is responsible for covalent linkage between PAPP-A and STC2. Inset shows map around the C120-C732 disulfide bond. The map (MAP1) was contoured at 2.0 σ. **d** Assessment of the ability of wild-type PAPP-A and PAPP-A(C732) to form a covalent complex with STC2. After incubation (16 h), the separately synthesized proteins were subjected to non-reducing SDS-PAGE and visualized by Western blotting for PAPP-A. The PAPP-A(C732) variant is unable to form the covalent PAPP-

A·STC2 complex with STC2. The experiment was repeated three times with similar results. Source data are provided as a Source data file. **e** Cartoon representation of the PAPP-A dimerization region within the PAPP-A·STC2 complex, showing domains M5-6 and SCR1 with the two PAPP-A subunits colored differently (cyan and transparent blue). The enlarged view emphasizes the dimerization interface and the disulfide bond between the two PAPP-A subunits. Note that the left M5-6 domains continue in the right SCR1 domain and vice versa. **f** Cartoon representation of the 10 SCR domains (shades of blue) of the PAPP-A·STC2 complex shown in two orientations (SCR1 (C1215-V1283), SCR2 (D1284-L1344), SCR3 (M1345-V1413), SCR4 (T1414-M1474), and SCR5 (Q1475-C1554)). **g** Top view of the PAPP-A·STC2 complex with only the M domains (cyan) and the SCR domains (blue) shown to emphasize their relative orientation. The lines drawn indicate the approximate planes in which M1-4 and the SCR domains, respectively, lie. The dihedral angle between the planes is indicated.

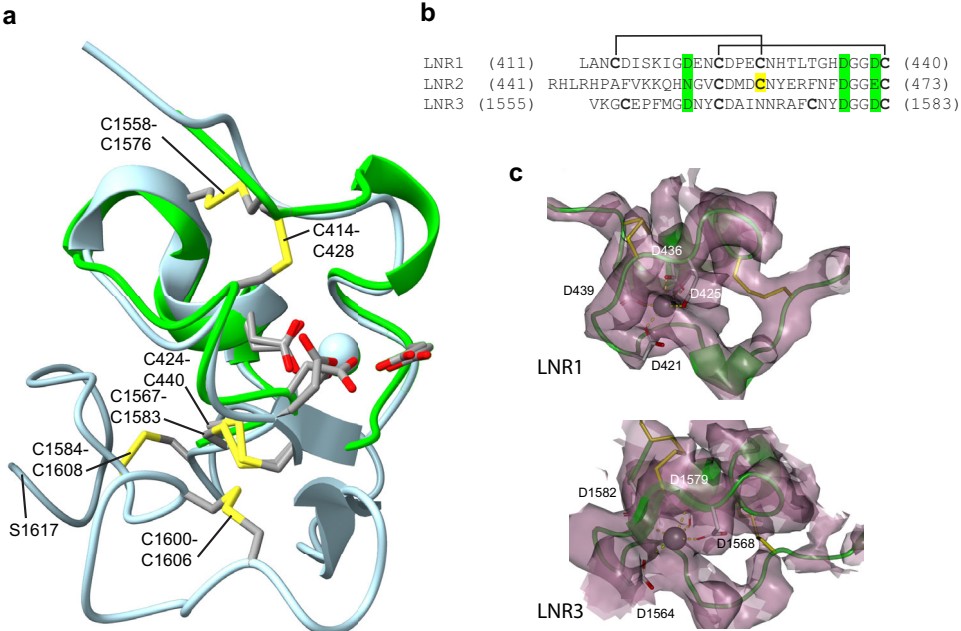

**Fig. 6 | The Lin12-Notch repeat (LNR) modules of the PAPP-A subunit. a** Cartoon representation of PAPP-A LNR1 (green) and the C domain (light blue), including LNR3 (dark green). LNR1 and LNR3 are superimposed. Disulfide bonds (yellow) and the position of the two Ca²⁺ ions (spheres) are shown. **b** Sequence alignment of the three PAPP-A LNR modules emphasizing coordinating residues (highlighted in green). LNR1 has two disulfide bonds. LNR2 has one disulfide bond and an unpaired cysteine (C461, marked yellow). LNR3 has two disulfide bonds arranged in the 1-3/2-4 pattern of LNR1, but two of the cysteine residues are not aligned. **c** Maps around the Ca²⁺ ions of LNR1 and LNR3, respectively. The map (MAP1) was contoured at 4.0 σ.

structure formed with LNR3, the C domain contains almost no secondary structure (Fig. 6a).

### The STC2 dimer

The STC2 dimer is positioned in the upper center of the PAPP-A·STC2 structure in the space between the two SCR5 domains with the dimerization disulfide (C211-C211) pointing upward (Fig. 7a). Each STC2 subunit contains a bundle of 11 antiparallel α-helices, several of which are oriented at an ≈90° angle relative to the corresponding helix of the other subunit, and connected with a short loop to the following helix (Fig. 7b and Supplementary Fig. 15). Each subunit is stabilized by six intrachain disulfide bonds (Supplementary Fig. 10), and the dimerization interface is formed by residues of the three helices between E169 and C211 (Fig. 7b). The density map does not allow structural modeling corresponding to sequence after C211 or before R44, suggesting that these regions are flexible. A search for structural homologs did not reveal other proteins with similar fold. Thus, the structure of STC2 presented here is the first of its family.

The STC2 dimer is suspended in the core of the PAPP-A·STC2 complex. In addition to the disulfide bond formed with the M2 domain (Fig. 5c), the STC2 dimer interacts noncovalently with the C domain (Supplementary Fig. 16), including electrostatic interactions between the C domain and four basic residues of STC2, and via van der Waal interactions between STC2 V63 and a hydrophobic pocket formed by PAPP-A residues Y1566, T1594, and K1592 (Fig. 7c, Supplementary Fig. 17). In particular, K104 is positioned favorably for interaction with the negative charge surrounding the Ca²⁺ ion of LNR3. These regions of PAPP-A-STC2 interactions appear generally to be highly conserved between species (Supplementary Fig. 18).

To assess the functional strength of such interactions, we determined the inhibitory properties of the STC2(C120A) variant. Although this variant cannot bind irreversibly to PAPP-A, we find that it is still a relatively potent competitive inhibitor (Fig. 8a).

C-terminally truncated variants of PAPP-A, PAPP-A lacking LNR3, or PAPP-A in which a Ca²⁺-coordinating residue of LNR3 is mutated to alanine, cannot form a covalent complex with STC2. In contrast, in the absence of LNR1-2, STC2 is still able to form the PAPP-A·STC2 complex with PAPP-A (Fig. 8b). We also find that a monoclonal antibody, PA141, which binds the PAPP-A C domain with high affinity to inhibit cleavage of IGFBP-4[36], cannot bind PAPP-A·STC2 (Fig. 8c). This antibody also effectively prevents complex formation (Fig. 8d). Together, these data suggest that the C domains are critically involved in the formation of the PAPP-A·STC2 complex.

Furthermore, we find that STC2 is able to bind an isolated monomeric C-terminal fragment of PAPP-A containing the C domain, and that this interaction can be abrogated by the PA141 antibody (Fig. 8e), thus suggesting that STC2 and PA141 have overlapping binding sites in the PAPP-A C domain, and that the inhibitory mechanism of mAb PA141 is based on mimicking of the endogenous inhibitor, STC2. Removal of bound Ca²⁺ from LNR3 of the C-terminal fragment did not completely abolish STC2 binding (Fig. 8f), indicating that residues C-terminally to LNR3 are likely to be involved in the binding. Finally, we similarly find that the substrate, IGFBP-4, is capable of binding to the isolated C-terminal fragment, and that this interaction is diminished by chelation of Ca²⁺ (Fig. 8g), demonstrating that LNR3 is involved in the binding of both inhibitor and substrate.

### Discussion

We have determined the structure of the metalloproteinase, PAPP-A, in complex with its endogenous inhibitor, STC2, to a resolution of 3.1 Å. Some regions, in particular some of the SCR domains (4–7 Å, Supplementary Fig. 2e) and M5-M6 (5–8 Å, Supplementary Fig. 2e), were determined to a lower resolution, and will benefit from further structural analysis. The 500 kDa 2:2 complex is composed of interacting homodimers, stabilized by many intra- and interchain disulfide bonds, several interdomain contacts, and several bound Ca²⁺ ions. This represents the first structure of PAPP-A, STC2, or any fragments hereof.

Except for the established role of the SCR3-4 domains in cooperating to bind cell surface GAG[16], and the role further defined here for

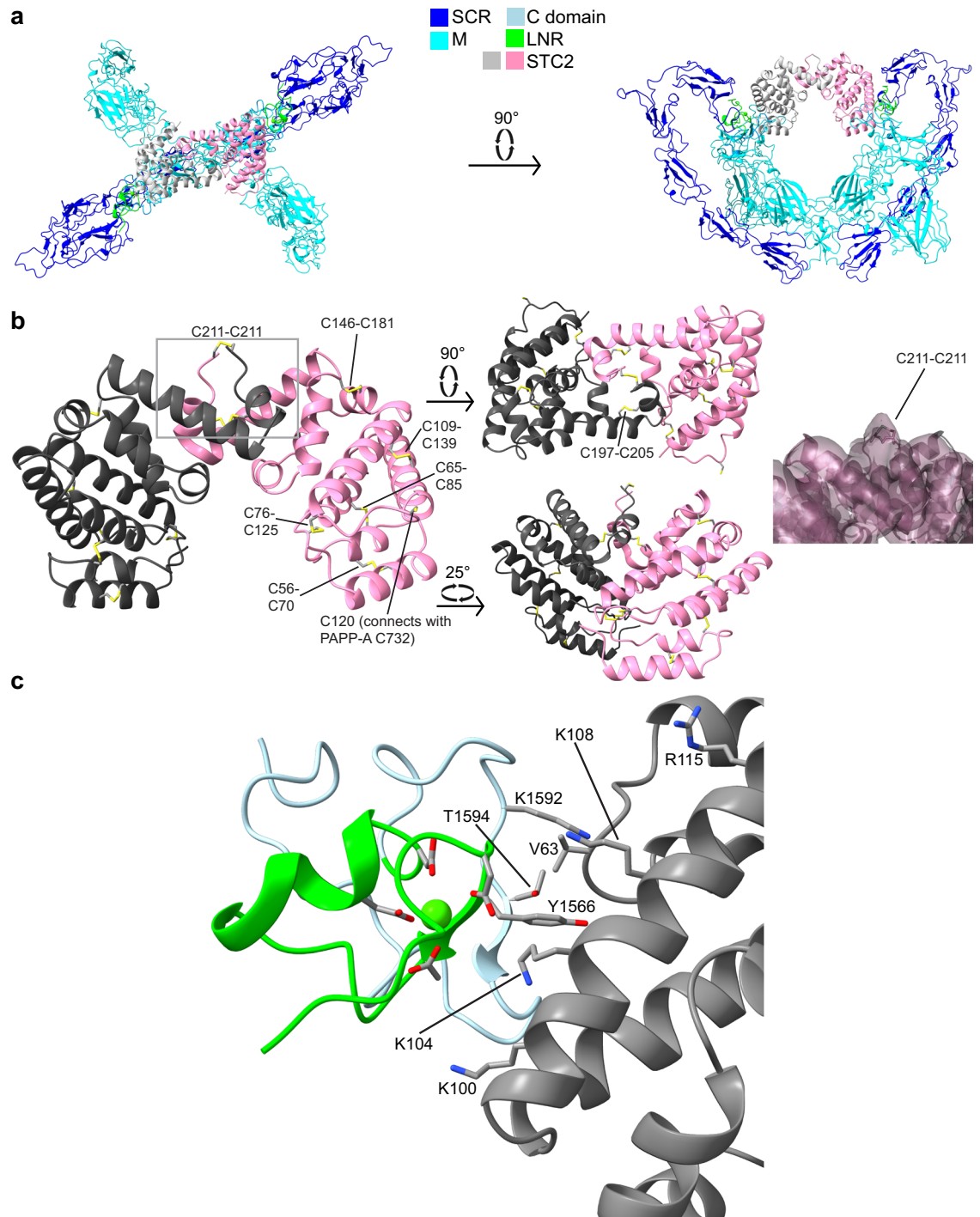

**Fig. 7 | The stanniocalcin-2 (STC2) dimer is suspended centrally in the PAPP-A·STC2 structure. a** Cartoon representation in two orientations of the PAPP-A·STC2 complex with the M domains (cyan), the SCR domains (blue), the C domain (light blue) including LNR3 (green), and the STC2 dimer (gray/pink) shown to emphasize the position of the STC2 dimer within the complex. This color scheme is used throughout the figure. **b** Cartoon representation of the STC2 dimer (R44-C211) of the PAPP-A·STC2 complex in different orientations, as indicated, to emphasize the relative orientation of the α-helices. Disulfide bonds are indicated. The density map around the C211-C211 dimerization disulfide bond is shown (right panel, position in the structure is indicated with a rectangle on the left panel). The map (composite map) was contoured at 7 σ. **c** The interface between STC2 and the PAPP-A C domain. Basic residues of STC2 positioned favorably for electrostatic interaction with the negative charge surrounding the $Ca^{2+}$ ion of LNR3 are indicated. Residues likely to be involved in van der Waal interactions are also indicated (V63 of STC2; Y1566, T1594, and K1592 of PAPP-A). Note that STC2 residues interact with the C domain of the PAPP-A subunit to which the opposite STC2 subunit is disulfide bound.

M5-6 in mediating PAPP-A dimerization, there is no biochemical data yet to suggest functional roles of the LG, M, and the SCR domains. In the PAPP-A·STC2 structure, the SCR3-4 domains make no contact with other domains and appear to be readily accessible for GAG interaction at the cell surface. The M domains appear as a scaffold, organizing the localization of the other domains.

While the core of the CD is similar in structure to other metzincins, in particular the two known unicellular pappalysin structures, the

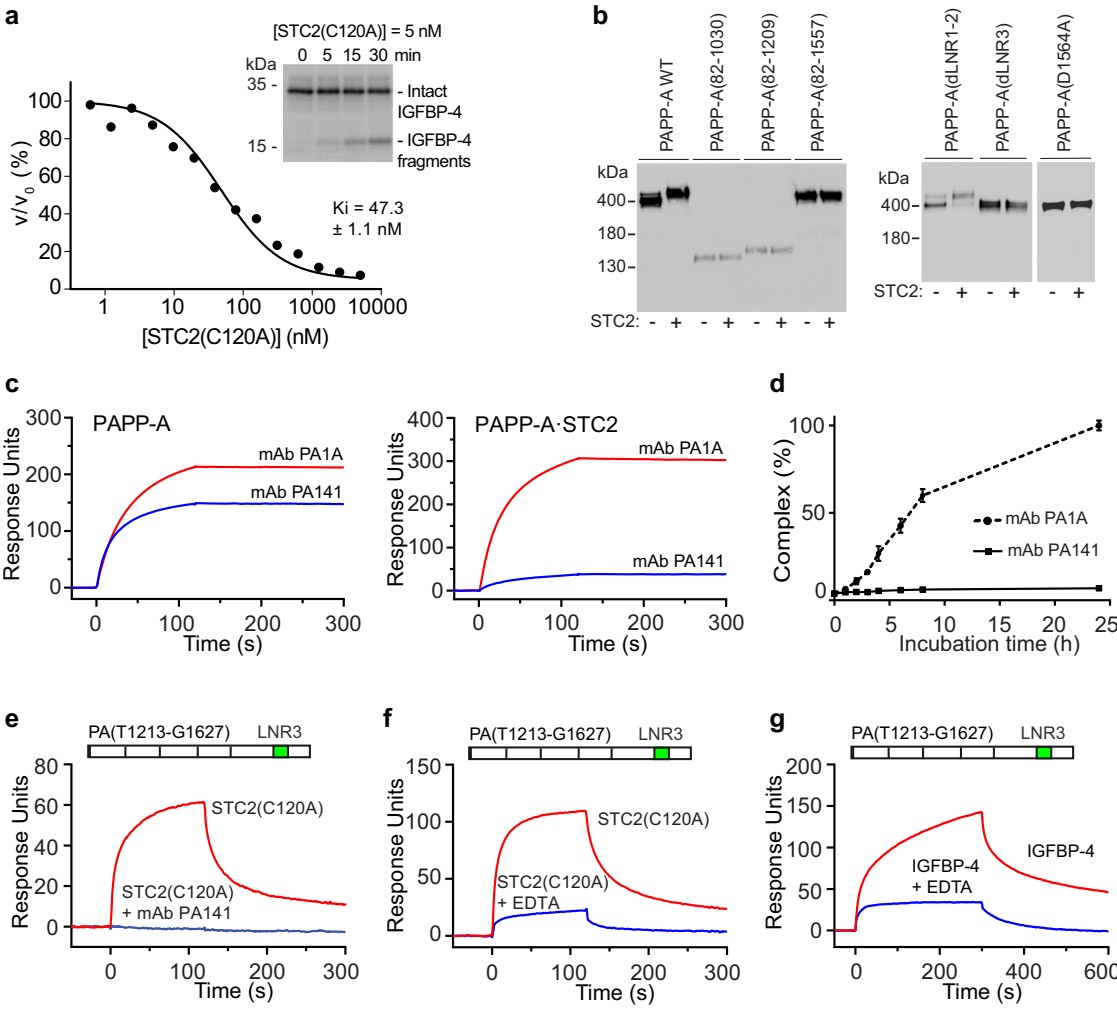

**Fig. 8 | Kinetics and interaction analyses. a** Kinetic analysis of inhibition of PAPP-A cleavage of IGFBP-4 by STC2(C120A). STC2(C120A) cannot bind covalently to PAPP-A. Intact radiolabeled IGFBP-4 was used, and relative initial velocities ($v/v_0$) of the cleavage reaction were determined by quantification of IGFBP-4 and cleavage products following separation by SDS-PAGE for each concentration of inhibitor (the inset shows an example). The concentrations of PAPP-A, IGFBP-4, and IGF2 were 50 pM, 10 nM, and 100 nM, respectively. The inhibition constant ($K_i$) was determined to be $47.3 \pm 1.1 \times 10^{-9}$ M by fitting the Morrison $K_i$ equation (competitive inhibition) to the data. The experiment was repeated three times with similar results. **b** Assessment of the ability of PAPP-A variants as indicated above the lanes to form a covalent complex with wild-type STC2. Following incubation (16 h) of separately synthesized proteins, the reaction mixtures were separated by non-reducing SDS-PAGE and visualized by Western blotting for PAPP-A. Only wild-type PAPP-A and PAPP-A(dLNR1-2) show increased molecular weight following incubation with STC2 and thus the ability to form a covalent complex. The experiment was repeated three times with similar results. **c** Binding of PAPP-A monoclonal

antibodies to PAPP-A (left) or PAPP-A·STC2 (right) assessed by surface plasmon resonance (SPR). The control antibody (mAb PA1A) binds both PAPP-A and PAPP-A·STC2, while the antibody mAb PA141, which inhibits PAPP-A activity toward IGFBP-4, shows diminished interaction with PAPP-A·STC2. **d** Complex formation between separately synthesized PAPP-A and STC2 over time assessed by an immunoassay specific for the PAPP-A·STC2 complex. PAPP-A mAb PA141 prevents complex formation. Average values with error bars (SD) are plotted. **e** Binding to a 415-residue monomeric C-terminal PAPP-A fragment, PA(T1213-G1627), assessed by SPR. PAPP-A mAb PA141 prevents binding of STC2(C120A). Average values with error bars (SD) are plotted. $n = 4$ independent experiments. **f** Similar to **e**, but carried out in the absence or presence of EDTA. The loss of bound $Ca^{2+}$ weakens the interactions between STC2 and the C domain. **g** Similar to **e**, but with IGFBP-4 as the analyte in the absence or presence of EDTA to disrupt binding of $Ca^{2+}$ to LNR3. The loss of bound $Ca^{2+}$ weakens the interactions between IGFBP-4 and the C domain. The sensorgrams of **c**, **e**, **f**, **g** are representative of at three independent experiments. Source data of **a**–**g** are provided as a Source data file.

PAPP-A CD stands out in several regards, most notable by the inserted LNR modules. Except for PAPP-A and PAPP-A2, the LNR module is present only in the family of Notch receptors. It is believed that the three tandemly arranged LNR modules of the Notch receptor shield a cleavage site within the receptor, and that pulling force generated by ligand binding causes its exposure, allows cleavage, and hence receptor signaling[37]. In PAPP-A, the LNR modules do not appear to have a similar role, but rather interact with either substrate or inhibitor.

The CD of PAPP-A is part of a large structure with the active site cleft oriented inward and close to domains that potentially interact with substrate. Curiously, cleavage of IGFBP-4 requires that it binds IGF[4], and accommodation of this IGFBP-4/IGF substrate complex of

~35 kDa at the active site is likely to involve interaction with neighboring domains, e.g., M2, in addition to its interaction with LNR3 as demonstrated here. However, it should be emphasized that the environment close to the active site cleft may differ in the PAPP-A dimer, in which there is no STC2 interaction and the restraints of the SCR modules are fewer than in the current heterotetrameric structure. A requirement for proteinase-substrate interactions is also illustrated by the finding that a 21-residue peptide derived from IGFBP-4 (A(123) KIRDRSTSGGKM-KVNGAPRE(143)) does not interact sufficiently with the PAPP-A dimer to become cleaved, although the 26-residue IGFBP-4-derived peptide used in this study (Fig. 4c), which has five more N-terminal residues, does[38].

The pappalysin S1' specificity pocket has a conserved aspartate in its bottom (D641 of PAPP-A, Fig. 4b and Supplementary Fig. 12b)[13,39] to accommodate the basic side chain of the substrate residue on the C-terminal side of the scissile bond, the P1' position[40]. The P1' position is occupied by a lysine residue in IGFBP-4 and -5. Curiously, in the absence of the C612-C643 disulfide bond, PAPP-A acquires autolytic activity, causing internal cleavage at sites with phenylalanine or tryptophan in the P1' position[12]. Thus, the C612-C643 disulfide bond, which connects the ZnM loop (V573-F635) underneath the active site cleft to the CD core, may be important for keeping D641 in a position to provide specificity. Several residues of the ZnM loop are necessary for cleavage of IGFBP-4[41] and therefore may also engage in substrate binding or contribute to stabilizing the ZnM loop. Altogether, intricate substrate interactions are supported by structural and biochemical data, but further analysis is required to understand, e.g., the unusual requirement that IGFBP-4 must bind IGF to become a substrate.

While IGFBP-5 can be cleaved by several different proteinases, IGFBP-4 is believed to be cleaved only by PAPP-A in vivo[1,8]. STC2 inhibits completely the latter proteolytic reaction in a highly unusual manner: (1) In the inhibited PAPP-A·STC2 complex we surprisingly find that the active site is still functional as shown by its ability to efficiently hydrolyze a 26-residue peptide derived from IGFBP-4. Thus, cleavage of unknown hypothetical PAPP-A substrates, e.g. smaller peptides, may or may not be affected by the binding of STC2 to PAPP-A. Based on the current data, we are unable to conclude whether STC2 causes restricted substrate access or reduces PAPP-A plasticity needed for substrate binding for the particular peptide substrate analyzed. Importantly, the mode of STC2 inhibition is fundamentally different from that of the TIMPs, the archetypical inhibitors of many mammalian metzincins (e.g., the MMPs and ADAMs), which function by active site cleft binding[42]. Curiously, PAPP-A inhibited by proMBP cannot cleave the used 26-residue peptide[43]. (2) Structural and biochemical data show that STC2 binds to the PAPP-A C domain, and that removal of Ca²⁺ from LNR3 diminishes its binding. We demonstrate that IGFBP-4 has an overlapping binding site in the C domain, consequently defining this region as a substrate-binding exosite, and STC2 as an exosite inhibitor. The previously identified inhibitory monoclonal antibody, PA1/41[36], also binds to this region, and therefore mimics the mechanism of the endogenous inhibitor. (3) C120 of STC2, which is required for the function of STC2 in vivo[21], connects specifically to C732 of the M2 domain of PAPP-A, thus preventing dissociation of the complex. The same cysteine residue of PAPP-A is also involved in covalent complex formation with proMBP[44,45], thus preventing binding of both inhibitors to PAPP-A at the same time. (4) In the absence of the STC2(C120)-PAPP-A(C732) disulfide, STC2(C120A) is still a functional competitive inhibitor similar to STC1 ($K_i = 68$ pM[20]), although relative to STC1, its efficiency is reduced ~700-fold. In summary, STC2 inhibits PAPP-A cleavage of IGFBP-4 irreversibly. Its mechanism of action is based on blocking access to the active site cleft of PAPP-A, and on binding to a substrate-binding exosite of the C domain.

Within the PAPP-A·STC2 complex, the C domains and the LNR3 modules of each PAPP-A subunit do not interact with the STC2 subunit to which the C120-C732 disulfide is formed, but rather the opposite subunit of the STC2 dimer. This reflects the crossing of the PAPP-A polypeptides at the PAPP-A dimerization disulfide, between M6 and SCR1 (Fig. 9a). Biochemical data have shown that PAPP-A subunits truncated before LNR3 are able to form homodimers, but such dimers are devoid of activity toward IGFBP-4[18] (Fig. 9b). However, this truncated variant is also able to dimerize with a C-terminal variant starting at M5, and such heterodimers are active toward IGFBP-4[18] (Fig. 9c). These data underscore that the M5 and M6 domains, here identified as responsible for formation of the dimerization interface, are sufficient for functional dimerization, and that the PAPP-A dimer has a trans configuration, which is maintained upon formation of the PAPP-A·STC2 complex.

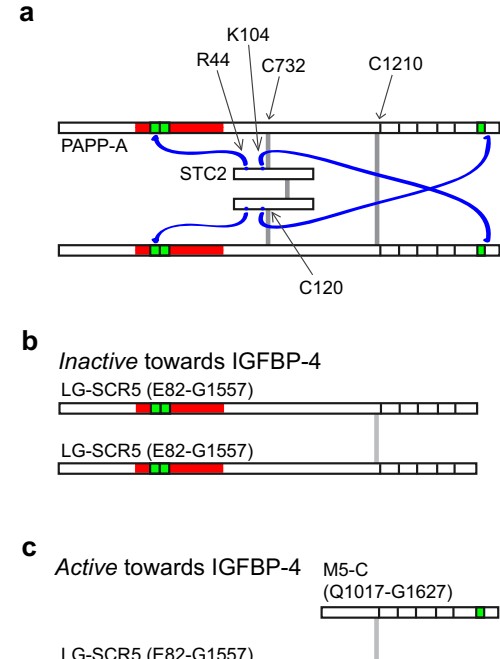

**Fig. 9 | Trans dimerization of the PAPP-A dimer is maintained in the PAPP-A·STC2 structure. a** Minimalistic schematic drawing of the PAPP-A·STC2 2:2 complex with indication of the intersubunit disulfide bonds (vertical gray bars) and observed (C domain-STC2) and hypothesized (LNR1-STC2) interactions (blue lines). The LNR modules are shown in green. Note that because the PAPP-A polypeptides cross at M6, LNR1 and LNR3 of each PAPP-A subunit do not interact with the same STC2 subunit. **b** Biochemical experiments have shown that PAPP-A subunits truncated before LNR3 are able to form dimers, but such dimers cannot cleave IGFBP-4[18]. **c** Heterodimerization between the same subunit truncated before LNR3 and a C-terminal fragment of PAPP-A, which is truncated before M5, rescues the proteolytic activity toward IGFBP-4[18], thus suggesting that PAPP-A is configured in trans before it forms the complex with STC2.

Recent genetic data illustrate the potential of regulated interactions between PAPP-A and STC2. For instance, single amino acid variants of STC2 were found to have a dramatic effect on human adult height, exemplified by the R44L mutation, which shows reduced ability of PAPP-A·STC2 complex formation and has an extreme height-increasing effect of 2.1 cm[25]. The PAPP-A·STC2 structure may help explain this observation, as the absence of an arginine in position 44 of STC2 prevents potential electrostatic interaction with LNR1-2. In the PAPP-A·STC2 complex, R44 is not positioned optimally for interaction with LNR1 or LNR2, but we hypothesize that in its absence, docking of STC2 into the PAPP-A dimer prior to disulfide bond formation is compromised.

It is established that formation of the PAPP-A·STC2 complex occurs post secretion[21]. Like other proteolytic systems, inhibition of PAPP-A by STC2 is subject to common mechanisms such as transcriptional regulation[46], but the trapping of PAPP-A by irreversible binding to STC2 may offer an additional level of regulation. The covalent PAPP-A·STC2 complex forms slowly over hours in vitro[21]. However, formation of the specific C732-C120 disulfide between PAPP-A and STC2 may potentially be subject to redox regulation in vivo, thus, e.g., promoting or preventing its formation.

Apart from growth physiology, the involvement of PAPP-A in the pathogenesis of multiple diseases has been established recently, e.g., cancer[47–51], atherosclerosis[23,52], kidney disease[24,53], and fibrosis[54]. The etiology of many of such age-related diseases has long been thought to involve IGF signaling[55], but targeting of the IGF receptor (IGF1R) has proven problematic with unwanted side effects[56,57]. Thus, targeting of

PAPP-A to locally decrease IGF signaling has the potential to circumvent this problem[8]. It is therefore important that in addition to providing a foundation for understanding the dynamic regulatory interactions of STC2 and PAPP-A within tissues, our finding open novel possibilities for therapeutic manipulation of the proteolytic part of the STC2 → PAPP-A → IGFBP → IGF axis.

## Methods

### Protein preparation

Recombinant proteins were expressed in human embryonic kidney 293T cells (293tsA1609neo, ATCC) cultured in DMEM (Thermo Fisher) supplemented with 10% fetal bovine serum (Sigma-Aldrich) or serum-free CD 293 Medium (Thermo Fisher). Cells stably expressing active site-inactivated human PAPP-A (293T_PA_E563Q_Hygro) were previously reported[12,22]. The construct covers the mature human PAPP-A sequence (E82-G1627) preceded by an artificial signal peptide[22]. Cells stably expressing human STC2 (293T_STC2_Hygro) were generated by transfection of 293T cells with linearized pSTC2_Hygro, obtained by cloning the XhoI/HindIII fragment of pSTC2[21] into pcDNA3.1/Hygro(+) (Invitrogen). The construct covers the full human STC2 sequence, including the native signal peptide (M1-R302). Transfected cells were cultured with hygromycin B (Invitrogen) and stably expressing clones were selected. To generate the PAPP-A·STC2 complex, 293T_PA_E563Q_Hygro cells and 293T_STC2_Hygro cells were mixed in a 1:1 ratio and co-cultured in serum-free medium. All other proteins were produced by transient transfection of 293T cells using calcium phosphate co-precipitation and plasmids encoding wild-type human PAPP-A (pcDNA3.1-PAPP-A[22], a C-terminal fragment of PAPP-A truncated at SCR1 (PA_T1213-G1627[18]), a PAPP-A variant with C732 replaced by alanine (PA_C732A[45]), a PAPP-A variant with D1564 replaced by alanine (PA_D1564A[19]), C-terminally truncated variants of PAPP-A (PA_E82-S1030, PA_E82-A1209, PA_E82-G1557[19]), deletion variants of PAPP-A (PA_d[C414−C473] and PA_d[C1558−C1583][19]), an STC2 variant with C120 replaced by alanine (pSTC2(C120A)[21]), or human IGFBP-4 (pBP4mH[12]). Culture supernatants were harvested 48 h post-transfection and cleared by centrifugation, or the cells were further cultured in serum-free medium to facilitate purification.

PAPP-A(T1213-G1627) and dimeric PAPP-A were purified by nickel-affinity and heparin-affinity chromatography[58], and IGFBP-4 was purified by nickel-affinity, size exclusion, and reversed-phase chromatography[4], as previously described. The PAPP-A·STC2 complex for cryo-EM analysis was purified by nickel-affinity chromatography using a 1 mL HisTrap HP column (Sigma-Aldrich). Serum-free culture supernatant was diluted 1:1 with 50 mM $NaH_2PO_4$, 150 mM NaCl, pH 7.4 (buffer A) and loaded onto the column, which was then washed with 50 mM $NaH_2PO_4$, 1 M NaCl, 20 mM imidazole, 0.05% Tween-20, pH 7.4, equilibrated with buffer A and eluted in 1 mL fractions with 50 mM $NaH_2PO_4$, 500 mM imidazole, pH 7.4. Following dialysis into 20 mM HEPES, 100 mM NaCl, 1 mM $CaCl_2$, pH 7.4 (buffer B), the eluate was passed through a 1 mL HiTrap NHS-activated HP column (Sigma-Aldrich) coupled with mAb PA141[36], which specifically recognizes dimeric PAPP-A, but not PAPP-A·STC2 (a finding of this study). The unbound PAPP-A·STC2 was finally concentrated by ultrafiltration using Amicon Ultracel 100 K (Millipore) and further purified by size exclusion chromatography on a Superose 6 Increase 3.2/300 GL (Sigma-Aldrich) equilibrated with buffer B. Throughout, fractions were analyzed by 3−8% non-reducing or reducing SDS-PAGE (NuPAGE, Thermo Fisher). To assess the integrity of the purified PAPP-A·STC2 complex, digests using trypsin (Supplementary Data 1) or endoproteinase Glu-C (V8 protease) were analyzed by mass spectrometry. Briefly, samples were denatured by 8 M urea, and reduced and alkylated by DTT and iodoacetamide. The samples were treated with trypsin, and the tryptic peptides were isolated by micropurification. LC-MS/MS was performed on an EASY-nLC 1200 system (Thermo Scientific) connected to an Orbitrap Eclipse Tribrid Mass Spectrometer

(Thermo Scientific). The data were searched against a database containing the sequences of PAPP-A and STC2 using the Sequest HT search engine. The search parameters were trypsin (semitryptic) as the protease, and a precursor mass tolerance of 10 ppm. Static modifications were set to carbamidomethyl and dynamic modifications with oxidation of methionine residues. The expected C-terminal of both subunits was confirmed, except that peptides representing D1620-G1627 of PAPP-A or E289-R302 of STC2 were not identified in the preparation used for cryo-EM (Supplementary Fig. 1b). The purified PAPP-A·STC2 complex was stored at 4 °C.

The sequence of constructs encoding wild-type PAPP-A or STC2 was in agreement with UniProt entries Q13219 and O76061, respectively. The PAPP-A sequence of some earlier databases and publications erroneously has a valine inserted between R107 and L108.

For preparation of murine monoclonal antibodies (PAPP-A mAbs 234-5, PA6, PA141, PA1A, and STC2 mAb STC221, see below), hybridoma clones were cultured in CD Hybridoma Medium (Thermo Fisher). For purification, culture medium was diluted 1:1 with 50 mM sodium phosphate, 150 mM NaCl, pH 7.4, and loaded onto a 1 mL HiTrap Protein G column (Cytvia). After washing with 50 mM sodium phosphate, 500 mM NaCl, 0.01% Tween-20, pH 7.4, bound antibodies were eluted with 100 mM glycine, pH 2.2. Fractions were neutralized with 1 M Tris, pH 9.0, diluted with PBS, and loaded onto a 1 mL Mono Q column (Cytvia). Bound antibodies were gradient eluted using PBS as buffer A, and PBS supplemented with 1 M NaCl as buffer B. Finally, collected fractions were dialyzed into PBS, and concentrations were determined by measurement of absorption at 280 nM. The purified antibodies were stored at 4 °C.

### Mass photometry

The molar masses of purified PAPP-A dimer and its covalent complex with STC2 were measured in 10 mM sodium phosphate, 140 mM NaCl, pH 7.4 (PBS) buffer using a Refeyn TwoMP mass photometer and the droplet dilution method. The sample concentrations were optimized empirically to give an event rate of ~2000 min$^{-1}$, which was achieved at ~10 nM. Scattering was converted to molar mass using a three-point calibration curve consisting of monomeric and dimeric BSA as well as thyroglobulin.

### Cryo-electron microscopy (cryo-EM)

Samples of PAPP-A·STC2 were diluted to a concentration of 0.6 mg/mL. C-flat holey carbon grids, CF-2.2-4C (Protochips), were glow-discharged on a Quorum GloQube Plus glow discharge system at 15 mA for 45 s. 3 μL of protein sample were added to the grids and vitrified at 4 °C and 100% humidity with a blotting time of 4 s on an EM GP2 automatic plunge freezer (Leica). Movies were collected on a Titan Krios G3i (EMBION, Danish National Cryo-EM Facility, Aarhus node) with X-FEG operated at 300 kV and equipped with a Gatan K3 camera and a Bioquantum energy filter using a 20 eV slit width. Movies were collected using aberration-free image shift (AFIS) data collection with the EPU data acquisition software (Thermo Fisher Scientific) at a pixel size of 0.507 Å/pixel (corresponding to a magnification of ×165,000). Two datasets were collected. Dataset I contained 10,073 movies with 59 dose fractions over a 0.8 s exposure and a total dose of ~58 e$^-$ per Å$^2$. Defocus range was 0.6–1.6 over six exposures in each hole. Dataset II contained 32,115 movies with 67 dose fractions over a 0.91 s exposure and a total dose of ~59 e$^-$ per Å$^2$. Defocus range was 0.6–1.8 over seven exposures in each hole). Further details are given in Supplementary Fig. 2 and Table 1.

### Data processing

Data processing was carried out by using cryoSPARC v3.3.2[59] based on datasets I and II. Three maps were generated. MAP1: Patch-motion-correction and Patch CTF determination was performed. Micrographs were manually inspected before initial manual particle picking.

Particles were used to create 2D classes for template picking on all movies, with a dynamic mask of 180–220 Å, followed by inspection of the picked particles. Particles were extracted in a 440-pixel box and Fourier cropped to a 128-pixel box (1.74 Å/pixel). Extracted particles (6,677,370) were used for iterative 2D classification, resulting in a set of particles used for ab initio 3D construction. Several rounds of heterogeneous refinement were performed to sort out junk particles and keeping one good class. Selected particles (278,982) were re-extracted in a 220-pixel box (1.01 Å/pixel) and subjected to non-uniform refinement followed by local refinement. MAP2: Initially, particles were picked using templates made from dataset I. After several rounds of 2D classification, six classes comprising ~3000 particles were used for Topaz training[60]. Topaz picking resulted in 601,179 particles. After particle clean-up with 2D classification, five initial models were built and heterogeneously refined. One model was refined with imposed C2 symmetry. In order to enhance the resolution, two more rounds of heterogenous, homogenous, and non-uniform refinement were performed (the final number of particles was 114,226). MAP3: Particles from the MAP2 were subjected to C2 symmetry expansion that was used with MAP2 volume for 3D variability analysis in C1 symmetry. Four conformations identified were used as seeds in heterogenous refinement. Refinement in C1 was performed on all four conformations, yielding volumes with clearly distinct conformations, and the best one was picked as MAP3 (the final number of particles was 31,119). A composite map (MAP1-3) was generated by using PHENIX v1.20.1[61]. Further details are given in Supplementary Fig. 2 and Table 1.

## Model construction

Initial PAPP-A (UniProt Q13219) and STC2 (UniProt O76061) models were calculated separately using AlphaFold2 (AF2)[28]. A partial dimer model was created from the LG, CD, M1-4, SCR2-SCR3, SCR5, C, and STC2 domains from the AF2 models, encompassing domains LG, CD, M1-4, and SCR2 that appear to move as a rigid body in Supplementary Movie 1, and domains SCR3, SCR5, and C, which also displayed side-chain densities. These models were placed in the 3.05 Å map (MAP1) and subjected to multiple cycles of manual adjustments and rebuilding in Coot v0.9.8.3[62] and real space refinement in PHENIX[63]. The model (deposited under the PDB ID 8A7D) contains chain C (PAPP-A S94-Y1014) from one PAPP-A subunit, chain Q (PAPP-A P1282-P1412 + PAPP-A G1476-S1617 from the other PAPP-A subunit, and chain P corresponding to one STC2 subunit (STC2 R44-F210). The local resolution varies significantly in MAP1 (Supplementary Fig. 7b), hence, the model side-chain positions of the ZnM loop, SCR3, and SCR5 should be taken with caution. Thus, PAPP-A regions 1015-1281 (M5, M6, SCR1) and 1413-1475 (SCR4), and STC2 regions 22-43 and 210-302 were omitted due to poor map coverage and pronounced flexibility in the SCR region. Subsequently, a full PAPP-A·STC2 heterotetrameric 2:2 model (deposited under the PDB ID 8A7E) was constructed by first docking two copies of the 8A7D structure into a PHENIX-calculated composite map of MAP2 and MAP3 using ChimeraX 1.3[64] followed by refinement as two rigid bodies using PHENIX. Next, AF2 Multimer was used to calculate a dimeric M5-M6-SCR1 model that was docked into the same map and subsequently refined using Namdinator[65], as especially the positions of the two 2-strand β-sheets of M6 needed significant adjustments. Even though the resolution of the composite map is low in M6, the main chain is readily traceable and clearly shows a crossover (which is not present in the AF2 model), i.e., SCR2 interacts with the LG domain of the adjacent PAPP-A monomer. Side chain positions, however, should be taken with caution in M6, and detailed structural characterization of this region would benefit from a future determination of the crystal structure of, e.g., a M5-M6-SCR1 dimer. The loop regions of M5 also displays low resolution. Two copies of the AF2 model of SCR4 were next manually docked into the composite density and refined as a rigid body. Due to the flexibility of PAPP-A, this is the least well-defined domain of the structure. The six fragments were then manually connected into two chains using Coot. An AF2 Multimer model of the STC2 dimer could readily be docked into the density and subsequently be manually rebuild and refined in PHENIX. Finally, the entire complex was refined with chains as rigid bodies using PHENIX. Further details are given in Table 1. Structural figures were prepared by using ChimeraX 1.3[64] or PyMOL v2.5.2 (https://www.pymol.org).

## Proteolytic assays and kinetic analysis

Cleavage of a 26-residue peptide derived from IGFBP-4 (L(118)QKHFAKIRDRSTSGGKMKVNGAPRE(143): position 131, o-aminobenzoic acid-modified lysine [Lys(Abz)]; position 139, 3-nitrotyrosine [Tyr(NO2)])) was assessed using an assay based on quenched fluorescence[43]. Reactions were carried out in 100 μL volumes in black Optiplate-96 F microplates (Perkin Elmer), and fluorescence cleavage signals (410 nm emission, 310 nm excitation) were measured over time using an EnSpire Multimode Plate Reader (PerkinElmer). Concentrations of PAPP-A or PAPP-A·STC2 were 1 nM. Initial cleavage velocities were calculated and plotted as a function of peptide concentration (1.25–40 μM). The Michaelis-Menten equation was fitted to the data and plotted for PAPP-A ($R2 = 0.97$) and PAPP-A·STC2 ($R2 = 0.94$). Determination of inhibitory potency of STC2(C120A) toward PAPP-A cleavage of intact IGFBP-4 was carried out essentially as described[20,43] for STC1 using radiolabeled substrate ($^{125}$I-IGFBP-4, 10 nM) under the assumption of competitive inhibition. The concentration of PAPP-A and IGF2 were 50 pM and 100 nM, respectively. Proteolytic reactions were initiated by the addition of preincubated $^{125}$I-IGFBP-4 and IGF2 (GroPep Bioreagents) in 50 mM Tris-HCl, 100 mM NaCl, 1 mM CaCl$_2$, pH 7.5. Following incubation at 37 °C, the reactions were terminated at various time points by the addition of hot SDS-PAGE sample buffer supplemented with 25 mM EDTA. Substrate and cleavage products were separated by 12% SDS-PAGE, and visualized by autoradiography using a storage phosphorscreen (GE Healthcare) and a Typhoon imaging system (GE Healthcare). Band intensities were quantified using the ImageQuant TL 8.1 software (GE Healthcare). Background levels (mock signals) were subtracted, and relative initial velocities (v/v$_0$) were determined by linear regression assuming no substrate depletion. Determinations of inhibition constants (K$_i$) were carried out by fitting the Morrison K$_i$ equation (competitive inhibition) to the data using the GraphPad Prism 5.0 software.

## Surface plasmon resonance (SPR)

SPR experiments were carried out using a Biacore 3000 instrument (Cytiva) running at 25 °C. The running buffer was 10 mM HEPES pH 7.5, 150 mM NaCl, 2 mM CaCl$_2$, and 0.05% Tween-20. In some experiments, the buffer was supplemented with 10 mM EDTA. Data were collected at a rate of 1 Hz, and analyzed using the BIAevaluation 4.1.1 software (Cytiva) and GraphPad Prism 9. Double referencing was applied, i.e. the signal from the in-line reference flow cell was subtracted, as was the signal from a blank run (0 nM analyte). For the analysis of interactions between monoclonal antibodies and PAPP-A(E563Q) or PAPP-A·STC2, an anti-PAPP-A chip was prepared by direct immobilization of antibody 234-5[66,67] to a level of 10,000 Response Units (RU) in flow cells 1 and 2 of a CM5 chip (Cytiva), which had been activated with a 1:1 mixture of 0.5 M 1-Ethyl-3-(3-dimethylaminopropyl) carbodiimide (EDC) and 0.1 M N-Hydroxysuccinimide (NHS). Remaining active groups were blocked by a 7 min injection of 1 M ethanolamine pH 8.5. For binding analysis, PAPP-A(E563Q) or PAPP-A·STC2 were captured at a level of ~600 RU in flow cell 2 only, using flow cell 1 as the in-line reference. Purified monoclonal PAPP-A antibodies (mAb PA1A[68] or mAb PA141[36]) were injected at a concentration of 100 nM for 2 min, at a 30 μl/min flow rate, followed by a 180 s dissociation phase. At the end of each cycle, the surfaces of both flow cells were regenerated by a 90 s injection of 10 mM glycine pH 2.2.

Interactions with PA_T1213-G1627 were analyzed in a similar manner. Briefly, PAPP-A mAb (PA6[69]) was immobilized on a CM5 chip

to a final response of 12,000 RU (active and reference cell), using standard amine chemistry as described above. PA_T1213-G1627 was then captured in one flow cell only (active flow cell), using an adjacent flow cell for in-line reference. Next, STC2_C120A was injected in both flow cells for 2 min at 10 µl/min. Regeneration between successive rounds was performed by a 2 min injection of 10 mM glycine pH 2.5. For the analysis of IGFBP-4 binding, PA_T1213-G1627 was immobilized on a CM5 chip (active cell only, using a blank cell as reference) to a level of 7000 RU (standard amine chemistry as above). IGFBP-4 was then injected in both reference and active flow cells for 5 min, at a 10 µl/min flow rate. Surfaces were regenerated by a 60 s injection of 10 mM glycine pH 1.7.

### Analysis of complex formation

To assess the ability to form a covalent complex, conditioned media containing PAPP-A or mutated variants were mixed (1:1) with wild-type STC2 and incubated for 16 h at 37 °C as described[21]. The mixtures were then separated by 3–8% non-reducing SDS-PAGE and analyzed by Western blotting. The separated proteins were blotted onto a PVDF membrane (Merck Millipore), and the membrane was blocked with 2% Tween-20 in 50 mM Tris-HCl, 500 mM NaCl, 0.1% Tween-20, pH 9.0 (TST), and incubated for 16 h at 22 °C with rabbit polyclonal anti(human PAPP-A) diluted 1:1000 in TST containing 2% skimmed milk powder[26,70]. The blot was then incubated at 22 °C with secondary swine polyclonal anti(rabbit IgG)-HRP (P0217, Dako), diluted 1:2000 in TST containing 2% skimmed milk powder for 1 h. TST was used for washing between all steps. Blots were developed using enhanced chemiluminescence (ECL Prime, GE Healthcare), and images were captured and analyzed using an ImageQuant LAS 4000 instrument (GE Healthcare). Uncropped images of blots are shown in the Source data file.

Alternatively, complex formation at 37 °C between PAPP-A (14 nM) and STC2 (29 nM) contained in serum media with or without mAb PA141 (20 nM) was monitored by a complex-specific time-resolved immunofluorometric assay (TRIFMA). OptiPlate-96 F HB black microplate wells (Perkin Elmer) were coated with 2 µg/mL PAPP-A mAb (PA6[69]) overnight at 4 °C, washed 3× with $H_2O$, blocked with 2% BSA in TBS for 1 h at RT, and washed 3× with TBS added 0.05% Tween-20 (TBS-Tw). Samples (100 µL of incubated mixtures) were diluted 22-fold in TBS-Tw added 1% BSA (TBS-Tw-BSA) and added to the wells. Following 1 h incubation at 37 °C, the wells were washed 3× in TBS-Tw and added 2 µg/ml biotinylated STC2 mAb (STC221[21]). Following incubation for 2 h at RT, the wells were washed 3×, added 1 µg/ml Europium-Streptavidin (Perkin Elmer), diluted with TBS-Tw added 25 µM EDTA, and incubated 1 h RT. Subsequently, the wells were washed and added 200 µl Enhancement Solution (Perkin Elmer). Time-resolved fluorescence was read on an EnSpire Multimode Plate Reader (Perkin Elmer).

### Reporting summary

Further information on research design is available in the Nature Research Reporting Summary linked to this article.

## Data availability

The cryo-EM 3D maps haves been deposited in the Electron Microscopy Data Bank (EMDB) with the following accession codes: EMD-15220 (MAP1), EMD-15217 (MAP2), EMD-15219 (MAP3), and EMD-15221 (composite map). The corresponding atomic models have been deposited in the Protein Data Bank (PDB) with the following accession codes: 8A7D (partial dimer model) and 8A7E (full heterotetrameric PAPP-A·STC2 complex). Source data are provided with this paper.

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

## Acknowledgements

This research was supported by grants to C.O. from the Independent Research Fund Denmark, the Carlsberg Foundation, and the Novo Nordisk Foundation, a grant to T.B. from the Danish National Research Foundation, a grant to Magnus Kjaergaard from the Carlsberg Foundation (mass photometry), and a grant to Jan J. Enghild from the Novo Nordisk Foundation (mass spectrometry). We would like to acknowledge Danish National Cryo-EM Facility (EMBION) for providing access to instruments, Jesper L. Karlsen for excellent support on matters related to scientific computing, and Pernille Noer for help with protein purification. We are grateful to Magnus Kjaergaard for excellent help with mass photometry, and to Jan J. Enghild and Carsten Scavenius for excellent help with mass spectrometry.

## Author contributions

S.D.K., T.B., and C.O. designed the research. S.D.K., S.K., T.R.K., and J.H.M. carried out the experiments. S.D.K., M.G., O.A.M., and T.B. performed cryo-EM data acquisition and processing, and built and refined cryo-EM structural models. C.O. wrote the manuscript with input and final approval from all authors.

## Competing interests

A patent application (inventors S.D.K., M.G., C.O.) relating to the structure has been filed (application number EP22192165.3). Otherwise, the authors declare no competing interests.
