## [Peer Review File · Nature Communications]

Structure of the proteolytic enzyme PAPP-A with the endogenous inhibitor stanniocalcin-2 reveals its inhibitory mechanismREVIEWER COMMENTS

Reviewer #1 (Remarks to the Author):

In this manuscript, SD Kobberø and colleagues report on a cryo-EM structure of the metalloprotease PAPP-A protein in complex with its endogenous inhibitor STC2 in attempt to delineate the inhibitory mechanism of PAPP-A. PAPP-P regulates IGF signaling by cleaving IGFBPs, thus presenting a potential target for intervening IGF signaling process. Lack of PAPP structure in free or in complex form has prevented us from understanding its molecular mechanism. The current paper directly addresses this problem by using cryo-EM structure analysis.

The study potentially addresses an interesting question regarding how PAPP-A is inhibited by STC2. If well validated as being correct, the current study would represent the first PAPP-A and STC2 structures, thus offering the first glimpse into such a molecular complex. However, there are several issues related to the presentation that may need to be addressed before its publication.

(1) There is lack of presentation of structure validation data. The authors need to show (i) the FSC calculated between the map and the final atomic model to verify the current half-map FSC resolution, (ii) the final atomic model fitting into the high-resolution component of the MAP1 to show how good the agreement between the final model and map is; (iii) the predicted model fitting into the low-resolution component of the maps to show the agreement of secondary structure elements that are supposed to be discernable at 5-6 Å; (iv) a supplementary table explaining which part or residues are solved at the atomic level and which part of the structure is modelled with pseudo-atomic model or backbone model. Please explicitly and briefly mention in the main text how the low-resolution components of the density maps are fitted and modelled.

(2) The current abstract could be misleading and may need revision. Although it is often not avoidable that cryo-EM maps contain a small percentage of low-resolution components, it becomes an issue when the major part or almost half of the map or the asymmetric unit is at low-resolution. In this case, right-out claiming 3-Å without putting clear definition of high-resolution component can be very much misleading. Please explicitly state in the main text that the structure is only partially solved at ~3-Å resolution if half of asymmetric unit of PAPP-A/STC2 is of low resolution, and clearly specify the exact percentage of the 3-Å component with respect to the complete structure.

(3) Because the authors used AlphaFold2 to generate the initial model, please provide a supplementary figure to show the confidence value-colored atomic model prediction of the monomeric PAPP-A and STC2, and also discuss how the low-confidence part of the structure is either replaced or corrected by atomic building into high-resolution part of the map or still corresponds to the low-resolution part of the map. For the latter, please put some caveats or warning that the atomic model is not or less reliable and belong to the area that requires attention in the future studies. One should not take it for granted that the low-confidence model fitting into low-resolution map means a solved structure, which is not acceptable and might well prove to be wrong latter when such components are eventually solved to atomic resolution.

(4) Can the authors draw a topology diagram for the new folds in STC2 (and likely also PAPP-A) not seen in other structures in the PDB database?

(5) Are the residues at the interface between PAPP-A and STC2 well resolved? If so, please show high-resolution local density map superimposed with the atomic model of the interface. If not, please provide experimental validation, such as mutagenesis to support the correctness of the residue assignment at the interface of PAPP-A and STC2.

(6) Please provide in a supplementary figure, the sequence alignment between different species to show the conservation of residues at the interface between PAPP-A and STC2.

(7) Please add a few more figure 3 panels to show the domain-colored monomeric model of PAPP-A and STC2 in different orientations. Please label M1-M6 in this figure.

(8) In figure 3, please provide at least two different orientations and locate the shown component in the entire complex.

(9) Please comment on if the reduced catalytic efficiency in the inhibited PAPP-P dimer is also true for its in vivo function and why only partial inhibition is required. If it is not clear if this results from artificial effects of purification rather than reflecting its function in cells, please add some caveats.

(10) On page 5, the phrase "2-fold symmetric focused density maps" is unclear. Do you mean that the maps are refined with imposing C2 symmetry? Then, what does "focused" mean? Which part of the complex is focused on in the refinement of these maps? Similar ambiguity is prevailed in the methods description.

In summary, this study is novel and important if adding necessary validation data or experiments. But the current writing and presentation is not rigorous enough to ensure its publication of the present form in Nature Communications. The authors need to considerably revise the paper not only by adding more data and figures but also practice structural rigorousness, particularly, put some necessary caveats and warning on the low-resolution and low-confidence part of the structure and not overclaim things that have not been done well.

Reviewer #2 (Remarks to the Author):

In this manuscript, Koberro et al. report the high resolution structure of the complex between the metzincin metalloproteinase PAPP-A and its endogenous inhibitor STC2, solved by state-of-the-art single particle cryoEM.

In addition, a series of functional assays were performed, notably by real-time surface plasmon resonance. These allowed to show that: 1) the antibody mAb PA141 inhibits the interaction between PAPP-A and STC2 by competing with STC2; 2) Cations (most likely Ca⁺⁺) are essential for the interaction between a 415-residue monomeric C-terminal PAPP-A fragment and both the C120A variant of STC2 (unable to form a covalent disulfide bond with the C732 residue of PAPP-A) and the substrate IGFBP-4, as both interactions are inhibited by the addition of EDTA.

The global experimental strategy is sound and the manuscript is clear and well written.

My only regret is that, beyond SDS-PAGE and mass spectrometry, no in solution method (such as SEC-MALS, mass photometry or analytical ultracentrifugation) was used to characterize the architecture of the PAPP-A/STC2 complex in parallel to cryoEM.

A very minor point: the abbreviation RU should be defined in the legend of figure 8.

Reviewer #3 (Remarks to the Author):

The present study on the "structure of PAPP-A in complex with its endogenous inhibitor STC2" by Kobbero, Gajhede et al. is an impressively well-executed study that builds on centuries of exceptional pappalysin research in the Oxvig Lab and perfectly utilizes AF2 generated models to finally unravel not only the structure of PAPP-A, but also the structure of its inhibitor STC2, and the complex thereof by CryoEM. Furthermore, they identified an intriguing exosite-driven inhibitory mechanism that prevents binding and, as a result, cleavage of IGFBP-4. It was an honor to review this work. The manuscript is well organized, concise, and to the point. As a result, I can enthusiastically recommend publishing in Nature Communications after minor revisions. Below some suggestions to further improve the manuscript.

Introduction:

- Please include some information about map resolution, model quality, etc. at the end of the introduction in the "summary" paragraph.

Results:

- Last paragraph on page 5: Please state the 8 experimentally verified glycosylation sites. And I am not sure if "substituted" is the best word to use here as it could be misleading. But I am not a native

speaker, so not sure.

- Please add an additional structural figure to the Supplement highlighting the experimentally identified glycosylation sites - if possible, in the same orientation as Figure 1C to aid orientation. Maybe also the putative sites could be added in a different color. But the focus should be on the ones seen in the structure.

- Page 6, top paragraph: Please add the domain boundaries of the "LG, "M" and "CD" here. I am aware they are stated in Figure 1, and at other places in the manuscript, but would be nice to have them also here to ease comprehension.

- Page 6, CD: Please indicate in the text the disulfides that were expected to be formed but couldn't verified by the structural model. And please state either here or in the discussion section if this may be to a low local map resolution, or if you believe that are simply not essential.

- Page 7, second paragraph: Please state the sequence of the 26-residue peptide used for activity monitoring. It is stated in the Figure Legend of Figure 4, but I would kindly ask to add this information also to the M&M section and the results.

- Page 7, second paragraph: Please discuss the observed putative μM -effect of PAPP-A inhibition by STC2 when using smaller substrates. Could STC-induced PAPP-A rigidity reduce plasticity needed for substrate binding at the active site? Or would the authors favor a model with reduced substrate access e.g.

- Is the effect of PAPP-A C732A and STC2 C120A identical in the activity assays and are similar K_i 's obtained? And can the authors estimate the K_i of wt-STC2? so e.g. is a solution of 100 μM PAPP-A fully inhibited by 100 μM STC2?

- Page 8, end of 2nd paragraph: The stated figure should be 5f and not 5e.

- Page 8, SCR domains: Why is it that the SCR2 is resolved best? Simply due to its contacts with the LG domain? Could it have derived from the cryoEM data analysis strategy which may defined it as part of the rigid body (chicken or egg dilemma)?

- Page 8, bottom: As "In the PAPP-A-STC2 complex, SCR1 is fixed and interacts with M5 in trans, SCR2 interacts with the LG domain in trans, and the C-terminal end of the PAPP-A subunit is also fixed (see below)." - I think it would be a good idea to provide an additional figure (in the same orientation as Fig. 1C) which focuses on the individual protein chains. For example by coloring the domains of one PAPP-A chain from the N- to the C-terminus going from "blue" to "red" or so, while keeping the other chain grey. And maybe, as the figure can be bigger than in 1C, you could add also the residue numbers of the PAPP-A domain boundaries in this figure.

- Page 9, top: Please add the putative biological roles of SCR3-SCR5 here.

- Page 9, LNR modules: I am confused about the second potential disulfide bridge in LNR3 (C1558-C1576) - Based on Supplementary Figure 3, it was experimentally not seen. But Figure 6a shows it clearly closed, while in 6b I am not sure if in the alignment it should be read "closed" or not. Please clarify. And what is the situation in your AF2 models - is it closed there? Or is that where the discrepancy is coming from?

- Page 9, bottom: You mention in the Methods section a proteomic verification of your recombinant proteins, but the data are not shown. Please show these data in a way so that the reader can judge also sequence coverage in an easy way.

- Page 10, bottom: Could be a good spot to drive home again the key point that both STC2 and PA141 seem to bind to the same or neighboring exosites, thereby preventing substrate binding while keeping the active site unaffected.

Discussion:

- Is it known where proMBP binds to PAPP-A? Is it a similar region? Would it be possible to form a complex of PAPP-A, STC2, and proMBP? Or does pro MBP protect PAPP-A from inhibition by STC2. Please discuss this potential interplay in the discussion.

- Page 11, top: Please indicate the lower resolution of the "full-length"-maps.

- Page 11, bottom: Do I understand it correctly that a 21-residue peptide is insufficient for cleavage by the PAPP-A dimer but the 26-residue version can be cleaved? Or is the 26mer only cleaved in the complex of the inhibitor? Would that open another route of regulation? So similar to IGFBP-4 requiring bound IGF to become a PAPP-A substrate, could it be possible that STC2 binding is needed to allow cleavage of certain smaller substrates?

- Page 13: The 700-fold reduction of inhibitory capacity refers to wt-STC2 or STC1 - please rephrase to make it unambiguous.

- Page 13, bottom: Do you have any biochemical data to corroborate the importance of R44 in STC2 for complex formation? e.g. by an inhibitory assay against IGFBP-4?

Methods:

- Please state the UniProt identifiers, potential sequence deviations, chosen sequence boundaries and the secretory signals (native signal peptide?) used for protein expression, and if the DNA was used as inferred from the corresponding cDNA or if it was codon optimized? I guess this information is available via other papers from the lab, but I strongly advocate to include this information also in this milestone publication to ease reproducibility.

- Page 15: Regarding the "inverse" purification using mAb PA141. Even if it is maybe a bit redundant, please quickly state that STC2-bound PAPP-A dimer is NOT recognized by this antibody. It is clear after reading the paper, but for people who are maybe mainly interested in the Methods, will be highly confused otherwise.

- Regarding the proteomic data from the Sciex TripleTOF 6600+ - Please show these data.

- Was there a special reason for using Topaz for particle picking? Low signal-to-noise in the recorded movies? Better 2D classes?

- Did the authors actually try to use Tristan Croll's Chimera X plugin ISOLDE to further improve their structural models? Maybe something to consider.

- Page 19, proteolytic assay - Please state the amino acid sequence of the 26mer.

Figures:

- Figure 1c: If possible, please indicate the active site of the protease in the figure. Or if not here, maybe in a supplementary figure.

- Figure 4b: The labels of T603 and E589 appear to be swapped in the upper-left figure.

- Figure 4c: Please discuss the k_M -driven inhibitory effect of STC2 against a peptidic substrate in the text. And please state in the figure that the IGFBP-4-derived 26mer was used for this analysis.
- Figure 6a: Why was LNR2 excluded from this analysis? Too different? Could it be shown side by side then? As stated also further up - I am a bit confused by the experimental status of disulfide C1558-C1576. And if it was seen, please improve Figure 6b.
- Figure 8a: Please add a similar SDS-PAGE-based insert for WT-STC2 to show full inhibition either at the same concentration or ideally, at 100x-lower protease/substrate concentration. Please also add the used PAPP-A concentration used for these assays in the figure legend.

Supplementary Figures:

- Please add the figure legend of Supplementary Figure 1 on all the individual pages of the figure, and the same, if applicable, for the subsequent figures.
- I would like to see a side-by-side comparison of the AlphaFold2 predicted PAPP-A with the experimentally determined structure (in the same orientation as used for Figure 2c), ideally actually for the multimer-prediction of the PAPP-A:STC2 complex.

Reviewer #1 (Remarks to the Author):

In this manuscript, SD Kobberø and colleagues report on a cryo-EM structure of the metalloprotease PAPP-A protein in complex with its endogenous inhibitor STC2 in attempt to delineate the inhibitory mechanism of PAPP-A. PAPP-P regulates IGF signaling by cleaving IGFs, thus presenting a potential target for intervening IGF signaling process. Lack of PAPP structure in free or in complex form has prevented us from understanding its molecular mechanism. The current paper directly addresses this problem by using cryo-EM structure analysis.

The study potentially addresses an interesting question regarding how PAPP-A is inhibited by STC2. If well validated as being correct, the current study would represent the first PAPP-A and STC2 structures, thus offering the first glimpse into such a molecular complex. However, there are several issues related to the presentation that may need to be addressed before its publication.

(1) There is lack of presentation of structure validation data. The authors need to show (i) the FSC calculated between the map and the final atomic model to verify the current half-map FSC resolution, (ii) the final atomic model fitting into the high-resolution component of the MAP1 to show how good the agreement between the final model and map is; (iii) the predicted model fitting into the low-resolution component of the maps to show the agreement of secondary structure elements that are supposed to be discernable at 5-6 Å; (iv) a supplementary table explaining which part or residues are solved at the atomic level and which part of the structure is modelled with pseudo-atomic model or backbone model. Please explicitly and briefly mention in the main text how the low-resolution components of the density maps are fitted and modelled.

OUR ANSWER: We agree with the reviewer about the appropriateness of including the requested material. We have expanded the first part of Results and divided it into two new paragraphs (pages 5 and 6), in which the raised validation questions (i, ii, iii, and iv) are addressed with reference to several new supplementary figures, as requested, and a table (Supplementary Fig. 3, 4, 5, and Supplementary Table 2). We had covered the fitting and modelling of the low resolution components in the experimental section in the previous version of the manuscript. We agree that the text was not sufficiently clear, and we have revised this paragraph (pages 20 and 21), and included an explanation of how the low-resolution components of the density maps were fitted and modelled, as requested.

(2) The current abstract could be misleading and may need revision. Although it is often not avoidable that cryo-EM maps contain a small percentage of low-resolution components, it becomes an issue when the major part or almost half of the map or the asymmetric unit is at low-resolution. In this case, right-out claiming 3-Å without putting clear definition of high-resolution component can be very much misleading. Please explicitly state in the main text that the structure is only partially solved at ~3-Å resolution if half of asymmetric unit of PAPP-A/STC2 is of low resolution, and clearly specify the exact percentage of the 3-Å component with respect to the complete structure.

OUR ANSWER: Again, we agree with the reviewer and appreciate the input. We have adjusted to abstract, now specifying the regions with 3.1 Å resolution. Regarding the request to specify in the main text: We have explicitly stated the requested information in the new, first two paragraphs of Results (cf. point 1).

(3) Because the authors used AlphaFold2 to generate the initial model, please provide a supplementary figure to show the confidence value-colored atomic model prediction of the

monomeric PAPP-A and STC2, and also discuss how the low-confidence part of the structure is either replaced or corrected by atomic building into high-resolution part of the map or still corresponds to the low-resolution part of the map. For the latter, please put some caveats or warning that the atomic model is not or less reliable and belong to the area that requires attention in the future studies. One should not take it for granted that the low-confidence model fitting into low-resolution map means a solved structure, which is not acceptable and might well prove to be wrong latter when such components are eventually solved to atomic resolution.

OUR ANSWER: We now provide the requested information in Supplementary Fig. 7a-h. Reference to these figures is also built into the first part of Results (cf. above). The comparison of the experimental local resolution and the AF confidence levels provided does not show immediately clear overall correlation. However, domains M5 and SCR4 have a clear correlation. This suggests to us that these domains are mediators of complex flexibility, as it is also known that the AF2 confidence level is an excellent predictor of disordered regions. We have included this in the first paragraph of Results. In the Methods (paragraph Model construction, pages 20 and 21), we have included caveats and warnings on the regions with low resolution, and we have also stated this uncertainty in the very first part of Discussion (page 12).

(4) Can the authors draw a topology diagram for the new folds in STC2 (and likely also PAPP-A) not seen in other structures in the PDB database?

OUR ANSWER: We have included a topology diagram for STC2 as Supplementary Fig. 15. Reference to the figure is made in Results, page 10. Although e.g. the inserted LNR modules make the CD unique compared to other metzincins, we feel that extended topology-based comparison for the PAPP-A domains would be beyond the scope and therefore we suggest not showing these. The novel M3 and M4 domains of PAPP-A are stabilized by disulfide bonds and calcium ion coordination, but do not contain any secondary structure.

(5) Are the residues at the interface between PAPP-A and STC2 well resolved? If so, please show high-resolution local density map superimposed with the atomic model of the interface. If not, please provide experimental validation, such as mutagenesis to support the correctness of the residue assignment at the interface of PAPP-A and STC2.

OUR ANSWER: We have prepared figures to illustrate the PAPP-A-STC2 interphases with the relevant map to show that it is well resolved, as requested (Supplementary Figure 17). Reference to this figure is made on page 11.

(6) Please provide in a supplementary figure, the sequence alignment between different species to show the conservation of residues at the interface between PAPP-A and STC2.

OUR ANSWER: We have included multiple sequence alignments (Supplementary Fig. 18) with interacting residues marked, as suggested. Reference to this figure is made on page 11.

(7) Please add a few more figure 3 panels to show the domain-colored monomeric model of PAPP-A and STC2 in different orientations. Please label M1-M6 in this figure.

OUR ANSWER (we assume that the comment is about Fig. 2): The suggested missing figure is clearly appropriate. However, we prefer to show after the identification of the M domains and the crossover of the two PAPP-A subunits. Thus, we have included the suggested figure (showing the PAPP-A monomer and the STC2 dimer) as Supplementary Fig. 16 with reference to it on page 11.

(8) In figure 3, please provide at least two different orientations and locate the shown component in the entire complex.

OUR ANSWER: We have revised Fig. 3 as requested.

(9) Please comment on if the reduced catalytic efficiency in the inhibited PAPP-P dimer is also true for its in vivo function and why only partial inhibition is required. If it is not clear if this results from artificial effects of purification rather than reflecting its function in cells, please add some caveats.

OUR ANSWER: The PAPP-A·STC2 complex is 0% active towards the full-length, physiological substrate IGFBP-4 (cf. ref 21). In Fig. 4c, we used a chemically modified 26-residue peptide derived from IGFBP-4 (thus an artificial substrate) only to demonstrate that the active site is accessible and not 'completely' inactivated towards other hypothetical, smaller physiological substrates. Such yet unknown substrates may or may not relate to the IGF system. The 26-residue peptide is modified at two positions (131, o-aminobenzoic acid-modified lysine, and 139, 3-nitrotyrosine) and therefore may interact 'artificially' (compared to native IGFBP4) with PAPP-A. For the point of demonstrating that the active site is functional, we could have used a qualitative peptide assay, but just demonstrating the presence of activity might have reflected perhaps >99% inhibition, which would be much less likely to be physiologically relevant. We chose to use the quantitative assay of Fig 4c to demonstrate that the active site of PAPP-A·STC2 is potentially available for smaller substrates, and that the activity is potentially only affected slightly (STC2 complex formation does not allosterically inactivate the active site). Thus, the observed reduction in catalytic efficiency may not be relevant to other, unknown substrates. Only after the identification of such substrates it will be possible to clarify.

(10) On page 5, the phrase "2-fold symmetric focused density maps" is unclear. Do you mean that the maps are refined with imposing C2 symmetry? Then, what does "focused" mean? Which part of the complex is focused on in the refinement of these maps? Similar ambiguity is prevailed in the methods description.

OUR ANSWER: We would like to thank the reviewer for pointing out these unclarities, especially the use of the word 'focused', which is not correct in this context. Thus, we have revised this wording which is a part of the first paragraph of Results (page 5).

In summary, this study is novel and important if adding necessary validation data or experiments. But the current writing and presentation is not rigorous enough to ensure its publication of the present form in Nature Communications. The authors need to considerably revise the paper not only by adding more data and figures but also practice structural rigorousness, particularly, put some necessary caveats and warning on the low-resolution and low-confidence part of the structure and not overclaim things that have not been done well.

Reviewer #2 (Remarks to the Author):

In this manuscript, Koberro et al. report the high resolution structure of the complex between the metzincin metalloproteinase PAPP-A and its endogenous inhibitor STC2, solved by state-of-the-art single particle cryoEM.

In addition, a series of functional assays were performed, notably by real-time surface plasmon resonance. These allowed to show that: 1) the antibody mAb PA141 inhibits the interaction between PAPP-A and STC2 by competing with STC2; 2) Cations (most likely Ca⁺⁺) are essential for the interaction between a 415-residue monomeric C-terminal PAPP-A fragment and both the C120A variant of STC2 (unable to form a covalent disulfide bond with the C732 residue of PAPP-A) and the substrate IGFBP-4, as both interactions are inhibited by the addition of EDTA.

The global experimental strategy is sound and the manuscript is clear and well written. My only regret is that, beyond SDS-PAGE and mass spectrometry, no in solution method (such as SEC-MALS, mass photometry or analytical ultracentrifugation) was used to characterize the architecture of the PAPP-A/STC2 complex in parallel to cryoEM.

OUR ANSWER: We agree that such alternative and independent support for the 2:2 stoichiometry is appropriate. We have estimated molecular weight of the PAPP-A dimer and the PAPP-A -STC2 complex by using mass photometry, as suggested, thus confirming the 2:2 stoichiometry of the complex (Supplementary Fig. 1a, made reference to in first paragraph of Results, page 5).

A very minor point: the abbreviation RU should be defined in the legend of figure 8.

OUR ANSWER: To avoid the abbreviation, we have chosen to spell out Response Units.

Reviewer #3 (Remarks to the Author):

The present study on the "structure of PAPP-A in complex with its endogenous inhibitor STC2" by Koberro, Gajhede et al. is an impressively well-executed study that builds on centuries of exceptional pappalysin research in the Oxvig Lab and perfectly utilizes AF2 generated models to finally unravel not only the structure of PAPP-A, but also the structure of its inhibitor STC2, and the complex thereof by CryoEM. Furthermore, they identified an intriguing exosite-driven inhibitory mechanism that prevents binding and, as a result, cleavage of IGFBP-4. It was an honor to review this work. The manuscript is well organized, concise, and to the point. As a result, I can enthusiastically recommend publishing in Nature Communications after minor revisions. Below some suggestions to further improve the manuscript.

Introduction:

- Please include some information about map resolution, model quality, etc. at the end of the introduction in the "summary" paragraph.

OUR ANSWER: We understand this request and agree. Rather than (briefly) stating in the summary in the end of the introduction we have (to avoid repetition) revised the first part of Results: We believe that the requested information is now clearly stated.

Results:

- Last paragraph on page 5: Please state the 8 experimentally verified glycosylation sites. And I am not sure if "substituted" is the best word to use here as it could be misleading. But I am not a native speaker, so not sure.

OUR ANSWER: We have stated the residues in the text. We agree that 'substituted' can be confusing and have changed to 'occupied' (page 6).

- Please add an additional structural figure to the Supplement highlighting the experimentally identified glycosylation sites - if possible, in the same orientation as Figure 1C to aid orientation. Maybe also the putative sites could be added in a different color. But the focus should be on the ones seen in the structure.

OUR ANSWER (we assume that the comment is about Fig. 2c): We have prepared the requested figure (Supplementary Fig. 11b) with the occupied asparagines (but not putative sites) highlighted and their identities pointed out (reference to on page 6).

- Page 6, top paragraph: Please add the domain boundaries of the "LG, "M" and "CD" here. I am aware they are stated in Figure 1, and at other places in the manuscript, but would be nice to have them also here to ease comprehension.

OUR ANSWER: We have added boundaries for CD and M, as requested, but not (repeated) boundaries for LG in this paragraph (now on page 7),

- Page 6, CD: Please indicate in the text the disulfides that were expected to be formed but couldn't verified by the structural model. And please state either here or in the discussion section if this may be to a low local map resolution, or if you believe that are simply not essential.

OUR ANSWER: We have added in the paragraph that we were unable to verify the C457-C473 disulfide bond due to low resolution (page 7).

- Page 7, second paragraph: Please state the sequence of the 26-residue peptide used for activity monitoring. It is stated in the Figure Legend of Figure 4, but I would kindly ask to add this information also to the M&M section and the results.

OUR ANSWER: We have added the amino acid sequence of the IGFBP4-derived peptide as requested (page 8).

- Page 7, second paragraph: Please discuss the observed putative k_M -effect of PAPP-A inhibition by STC2 when using smaller substrates. Could STC-induced PAPP-A rigidity reduce plasticity needed for substrate binding at the active site? Or would the authors favor a model with reduced substrate access e.g.

OUR ANSWER: We feel that we are unable to discriminate between the two, but have mentioned the possibilities as suggested in the discussion (page 14). We have used the words above ("reduce plasticity needed for substrate binding") in the additions made, and we would like to thank the

reviewer for the inspiration. Please also see our related answer below to points “Page 11, bottom” and “Figure 4c”.

- Is the effect of PAPP-A C732A and STC2 C120A identical in the activity assays and are similar K_i 's obtained? And can the authors estimate the K_i of wt-STC2? so e.g. is a solution of 100pM PAPP-A fully inhibited by 100pM STC2?

OUR ANSWER: We have not carried out the ‘reverse’ experiment using C732A for kinetic analysis, but chosen here one way of blocking the covalent interaction. We did this primarily to obtain a first glimpse of specific interaction between PAPP-A and STC2 in the absence of the possibility of covalent linkage. We picked C120A (over C732A) because we know that STC1 (in which C120 is missing) has high-affinity interactions with wild-type PAPP-A. In principle, interactions with neither C732A nor C120A may not 100% reflect the interactions we observe in the covalent ‘reaction product’. However, in an in-depth study aiming at understanding the process of complex formation (which we are currently planning separately) we believe that the comparison will be relevant along with several other biochemical studies to address the possible involvement of e.g. cysteinylolation, oxidation, and requirement for redox potential. (We hypothesize that the process of complex formation is potentially regulated (in vitro and in vivo) by redox potential). Regarding wt-STC2: A K_i value for wt-STC2 would have to be determined under artificial conditions where covalent complex formation does not occur, and it would thus only relate only to such conditions.

- Page 8, end of 2nd paragraph: The stated figure should be 5f and not 5e.

OUR ANSWER: Reference to ‘5e’ is actually correct. The relevant part of 5e is the right part. We have made the line indicating the enlarged view thicker (it was thin by mistake) to avoid confusion.

- Page 8, SCR domains: Why is it that the SCR2 is resolved best? Simply due to its contacts with the LG domain? Could it have derived from the cryoEM data analysis strategy which may defined it as part of the rigid body (chicken or egg dilemma)?

OUR ANSWER: In contrast to the other SCR domains, SCR2 has extensive protein-protein interactions with the M5 domain, and also as mention by the reviewer, the LG domain. The result is that LG, CD, M1-M5 and SCR2 apparently act as a rigid body in the dynamics of the complex in solution. This is derived from the 3D variability analysis undertaken and visualized in Supplementary Movie 1. We have adjusted to text (page 9) to reflect that this is our view.

- Page 8, bottom: As "In the PAPP-A-STC2 complex, SCR1 is fixed and interacts with M5 in trans, SCR2 interacts with the LG domain in trans, and the C-terminal end of the PAPP-A subunit is also fixed (see below)." - I think it would be a good idea to provide an additional figure (in the same orientation as Fig. 1C) which focuses on the individual protein chains. For example by coloring the domains of one PAPP-A chain from the N- to the C-terminus going from "blue" to "red" or so, while keeping the other chain grey. And maybe, as the figure can be bigger than in 1C, you could add also the residue numbers of the PAPP-A domain boundaries in this figure.

OUR ANSWER (we assume this point is about Fig. 2c): We agree and have included a figure (Supplementary Fig. 8) in which all of one PAPP-A subunit is colored light grey. Reference to the

figure is made on page 8. Together with Supplementary Fig. 16 (also new), we believe that the relevant points are now emphasized clearly.

- Page 9, top: Please add the putative biological roles of SCR3-SCR5 here.

OUR ANSWER: We have added (Page 10) the known role of domains SCR3-4 in mediating cell surface binding of PAPP-A to this paragraph.

- Page 9, LNR modules: I am confused about the second potential disulfide bridge in LNR3 (C1558-C1576) - Based on Supplementary Figure 3, it was experimentally not seen. But Figure 6a shows it clearly closed, while in 6b I am not sure if in the alignment it should be read "closed" or not. Please clarify. And what is the situation in your AF2 models - is it closed there? Or is that where the discrepancy is coming from?

OUR ANSWER: We apologize for this. Supplementary Fig. 3 was erroneous, and we have now corrected (thus, the C1558-C1576 disulfide was experimentally confirmed). Thanks for noticing. The alignment: We have adjusted the legend of Fig. 6b to specifically state that LNR3, like LNR1, has two disulfide bonds.

- Page 9, bottom: You mention in the Methods section a proteomic verification of your recombinant proteins, but the data are not shown. Please show these data in a way so that the reader can judge also sequence coverage in an easy way.

OUR ANSWER: We now show the identity of critical peptides in Supplementary Fig. 1b (and all tryptic peptides identified in Supplementary Table 3). This experiment was carried out on a different instrument because we planned (during revision) to evaluate stoichiometry from this experiment (another reviewer requested that). However, we also carried out mass photometry for accurate determination of molecular masses of PAPP-A and the complex (Supplementary Fig. 1a), and thus chose to not include a less accurate estimate based on calculations from mass spectrometry data. The new mass spectrometry data set, which is based on the protein preparation actually used for the final cryo-EM studies, lacked 8 C-terminal residues of the PAPP-A subunit and 13 C-terminal residues of the STC2 subunit, which we had previously observed using a different protein preparation. We believe it is most fair to report the data for the actual protein preparation, thus Supplementary Fig. 1b and Supplementary Table 3.

- Page 10, bottom: Could be a good spot to drive home again the key point that both STC2 and PA141 seem to bind to the same or neighboring exosites, thereby preventing substrate binding while keeping the active site unaffected.

OUR ANSWER: We have adjusted the text (page 11), as suggested, to emphasize this point.

Discussion:

- Is it known where proMBP binds to PAPP-A? Is it a similar region? Would it be possible to form a complex of PAPP-A, STC2, and proMBP? Or does pro MBP protect PAPP-A from inhibition by STC2. Please discuss this potential interplay in the discussion.

OUR ANSWER: Thanks - this is a good point. It is known that C732 is also involved in binding to proMBP, thus preventing simultaneous binding of both inhibitors. The PAPP-A-proMBP complex cannot cleave the 26-residue peptide used in this study. We have added both points to the discussion (page 14).

- Page 11, top: Please indicate the lower resolution of the "full-length"-maps.

OUR ANSWER: We have revised the first part of Results (pages 5 and 6) entirely to contain a detailed account of the maps used and their resolutions (and prepared several new supplementary figures that are made reference to in the first paragraph of Results).

- Page 11, bottom: Do I understand it correctly that a 21-residue peptide is insufficient for cleavage by the PAPP-A dimer but the 26-residue version can be cleaved? Or is the 26mer only cleaved in the complex of the inhibitor? Would that open another route of regulation? So similar to IGFBP-4 requiring bound IGF to become a PAPP-A substrate, could it be possible that STC2 binding is needed to allow cleavage of certain smaller substrates?

OUR ANSWER: Yes, the 21-residue peptide cannot be cleaved by PAPP-A, but the 26-residue can (cleavage occurs increasingly from 21 to 26 as more residues are added (cf. ref. 38). We selected the 26-residue peptide previously for an activity assay based on quenched fluorescence (cf. ref 43). There is no data to indicate that the PAPP-A-STC2 complex 'becomes active' towards smaller substrates (if the PAPP-A-STC2 had been more active than PAPP-A, we could have suggested that). No physiological 'small' substrates are known, but may exist as indicated in the text. If so, they may not be affected (or only slightly affected) by STC2 inhibition. We acknowledge the confusion and have tried to clarify (page13).

- Page 13: The 700-fold reduction of inhibitory capacity refers to wt-STC2 or STC1 - please rephrase to make it unambiguous.

OUR ANSWER: Thanks for pointing out. We have adjusted the text to be more precise about the comparison we intended to make. The point that wt-STC2 inhibits irreversibly is stated in the following sentence (page 14).

- Page 13, bottom: Do you have any biochemical data to corroborate the importance of R44 in STC2 for complex formation? e.g. by an inhibitory assay against IGFBP-4?

OUR ANSWER: Yes - thanks for mentioning that this was not clear. We have rearranged the words of this paragraph to clearly state that the mentioned R44L mutation with a dramatic effect on human height indeed forms the complex with PAPP-A more slowly (page 15).

Methods:

- Please state the UniProt identifiers, potential sequence deviations, chosen sequence boundaries and the secretory signals (native signal peptide?) used for protein expression, and if the DNA was used as inferred from the corresponding cDNA or if it was codon optimized? I guess this information is available via other papers from the lab, but I strongly advocate to include this information also in this milestone publication to ease reproducibility.

OUR ANSWER: UniProt IDs for PAPP-A and STC2 (Q13219 and O76061, respectively) and complete agreement between the cDNAs and the database sequences (except for the specified mutated variants) were already stated at the end of this paragraph. We have added sequence boundaries, as requested (Page 17). We have also added that for PAPP-A expression an artificial signal peptide was used and made a clear reference to the publication in which this was first used (page 17). For clarification, we have also added to the first paragraph of Results (page 6) and to the legend of Fig. 2 that PAPP-A(E563Q) was used for structural analysis. Information about variants in some earlier publications was already stated at the end of the paragraph.

- Page 15: Regarding the "inverse" purification using mAb PA141. Even if it is maybe a bit redundant, please quickly state that STC2-bound PAPP-A dimer is NOT recognized by this antibody. It is clear after reading the paper, but for people who are maybe mainly interested in the Methods, will be highly confused otherwise.

OUR ANSWER: We do understand this point. To clarify, we have changed the sentence to: "(...) which specifically recognizes dimeric PAPP-A, but not PAPP-A·STC2 (a finding of this study)" (page 18).

- Regarding the proteomic data from the Sciex TripleTOF 6600+ - Please show these data.

OUR ANSWER: Please see our answer to the relevant point above ("Page 9, bottom") (the data are in Supplementary Table 3).

- Was there a special reason for using Topaz for particle picking? Low signal-to-noise in the recorded movies? Better 2D classes?

OUR ANSWER: Several particle picking strategies have been attempted: Manual picking, manual picking followed by template based and Topaz picking based on selected templates. The Topaz based picking proved to give the largest particle sets with the highest resolution 2D classes, and for this reason these particles were used in the subsequent analysis.

- Did the authors actually try to use Tristan Croll's Chimera X plugin ISOLDE to further improve their structural models? Maybe something to consider.

OUR ANSWER: We have not used ISOLDE in the refinement of the structure. The refinement was carried out with the PHENIX package and COOT for manual rebuilding, an approach we are very experienced with. In addition, we have used Namdinator in difficult regions. Namdinator have essentially the same molecular dynamics functionality as ISOLDE, but does not have the manual rebuilding tools included. However, we agree that it would definitely be worth considering ISOLDE in future projects.

- Page 19, proteolytic assay - Please state the amino acid sequence of the 26mer.

OUR ANSWER: We have added (page 22) the sequence of the peptide derived from IGF1BP4, as requested.

Figures:

- Figure 1c: If possible, please indicate the active site of the protease in the figure. Or if not here, maybe in a supplementary figure.

OUR ANSWER (we assume that this is about Fig. 2c): We have prepared a separate figure (Supplementary Fig. 9) to emphasize this point. Reference to the figure is made on page 6.

- Figure 4b: The labels of T603 and E589 appear to be swapped in the upper-left figure.

OUR ANSWER: We have corrected this error. Thanks again for noticing!

- Figure 4c: Please discuss the K_M -driven inhibitory effect of STC2 against a peptidic substrate in the text. And please state in the figure that the IGFBP-4-derived 26mer was used for this analysis.

OUR ANSWER: Please see our comments to the point above (page 11, bottom). We would very much like not to discuss possible reasons for the K_M reduction for two reasons: First, the principle reason for the quenched fluorescence experiment is to demonstrate that the active site cleft is not occupied (and thus available to hypothetical physiological substrates, potentially to a variable degree. Some may be affected by STC2, some may not). Second, the peptide derived from IGFBP4 is modified at two positions (131, *o*-aminobenzoic acid-modified lysine, and 139, 3-nitrotyrosine) and thus may interact 'artificially'. For the point of demonstrating that the active site is functional, we could have used a qualitative peptide assay, but just demonstrating the presence of activity might have reflected perhaps >99% inhibition, which would be less likely to be physiologically relevant. We chose to use the quantitative assay of Fig. 4c to demonstrate that the active site of PAPP-A·STC2 is potentially available for smaller substrates, and that the activity is potentially only affected slightly (STC2 complex formation does not allosterically inactivate the active site). We have stated the identity of the 26-residue peptide in the legend of Fig. 4c.

- Figure 6a: Why was LNR2 excluded from this analysis? Too different? Could it be shown side by side then? As stated also further up - I am a bit confused by the experimental status of disulfide C1558-C1576. And if it was seen, please improve Figure 6b.

OUR ANSWER: Regarding the disulfide bonds, please see our answer above. LNR1 and LNR3 are more similar, both with two disulfides, however, we have included all three in comparison with Notch LNR (Supplementary Fig. 14).

- Figure 8a: Please add a similar SDS-PAGE-based insert for WT-STC2 to show full inhibition either at the same concentration or ideally, at 100x-lower protease/substrate concentration. Please also add the used PAPP-A concentration used for these assays in the figure legend.

OUR ANSWER: To avoid confusion (irreversible vs competitive), we would rather not include wild-type as the full inhibition would require pre-incubation for full complex formation (cf. ref. 21). We have added the requested information about concentrations to the legend to this figure.

Supplementary Figures:

- Please add the figure legend of Supplementary Figure 1 on all the individual pages of the figure, and the same, if applicable, for the subsequent figures.

OUR ANSWER: Thanks. We are aware that it requires some going back and forth between legend and figure, and we have added the legends of Supplementary Fig. 2 (the previous Suppl. Fig. 1), as suggested, and also in part for other supplementary figures. We believe that the remaining figures are either close to the legends already or self-explanatory (so copying legend onto the same page would potentially make figures smaller).

- I would like to see a side-by-side comparison of the AlphaFold2 predicted PAPP-A with the experimentally determined structure (in the same orientation as used for Figure 2c), ideally actually for the multimer-prediction of the PAPP-A:STC2 complex.

OUR ANSWER: We have generated the requested comparison, presented as part of Supplementary Fig. 7, made reference to in the second paragraph of Results. In this paragraph, we also comment of the resulting comparison (i.e. that attempts to predict the PAPP-A dimer (or the 2:2 complex) were not successful). We also point out that, e.g., “the most obvious difference between the PAPP-A monomer resulting from AF2 calculations and the PAPP-A subunit of the 2:2 experimental PAPP-A:STC2 complex is the markedly different spatial directions of the SCR modules”.

REVIEWERS' COMMENTS

Reviewer #1 (Remarks to the Author):

The authors have addressed most of the issues raised in my previous review. I'd like to recommend its publication given that that authors make the following suggested minor revision.

(1) On page 12, please clarify the resolution range for the second sentence of the Discussion section, that is, "Some regions, ..., were determined to a lower resolution, ...", and cite the directly related Supplementary figure(s) that can provide the information.

(2) Please move Supplementary Table 1 to main text Table 1. This table is critical to assess the quality of structure determination. The clashscore for PDB 8A7E seems to be somewhat less optimal and could be improved to a degree closer to the other PDB in the table.

Reviewer #3 (Remarks to the Author):

All of my concerns were addressed by the authors, who also made considerable changes to the manuscript. Thus I am happy to recommend that the work be published in Nature Communications.

Response to Reviewer 1 comments:

Reviewer #1 (Remarks to the Author):

The authors have addressed most of the issues raised in my previous review. I'd like to recommend its publication given that that authors make the following suggested minor revision.

(1) On page 12, please clarify the resolution range for the second sentence of the Discussion section, that is, "Some regions, ..., were determined to a lower resolution, ...", and cite the directly related Supplementary figure(s) that can provide the information.

We have added a reference (twice) to Supplementary Figure 2e, as requested.

(2) Please move Supplementary Table 1 to main text Table 1. This table is critical to access the quality of structure determination. The clashscore for PDB 8A7E seems to be somewhat less optimal and could be improved to a degree closer to the other PDB in the table.

We have deleted Supplementary Table 1 and now include the same information as "Table 1", as requested (and then renumbered Supplementary Tables 2 and 3 (thus, now Supplementary Tables 1 and 2). The clash core of the dimeric structure (8A7E) is compared to other available structures in the pdb validation report. It is higher than average, but not unusually high (low resolution structures do have higher clash scores than atomic resolution structures). We did focus on the clash score in the refinement of this structure, and thus ended at a clash score of 16, as reported (pdb validation).